# Allosteric modulation of LRRC8 channels by targeting their cytoplasmic domains

Dawid Deneka[1], Sonja Rutz[1], Cedric A. J. Hutter [2], Markus A. Seeger [2], Marta Sawicka [1]✉ & Raimund Dutzler [1]✉

Members of the LRRC8 family form heteromeric assemblies, which function as volume-regulated anion channels. These modular proteins consist of a transmembrane pore and cytoplasmic leucine-rich repeat (LRR) domains. Despite their known molecular architecture, the mechanism of activation and the role of the LRR domains in this process has remained elusive. Here we address this question by generating synthetic nanobodies, termed sybodies, which target the LRR domain of the obligatory subunit LRRC8A. We use these binders to investigate their interaction with homomeric LRRC8A channels by cryo-electron microscopy and the consequent effect on channel activation by electrophysiology. The five identified sybodies either inhibit or enhance activity by binding to distinct epitopes of the LRR domain, thereby altering channel conformations. In combination, our work provides a set of specific modulators of LRRC8 proteins and reveals the role of their cytoplasmic domains as regulators of channel activity by allosteric mechanisms.

[1] Department of Biochemistry University of Zurich, Winterthurerstrasse 190, CH-8057 Zurich, Switzerland. [2] Institute of Medical Microbiology University of Zurich, Gloriastrasse 28/30, CH-8006 Zurich, Switzerland. ✉email: m.sawicka@bioc.uzh.ch; dutzler@bioc.uzh.ch

In mammals, volume-regulated anion channels (VRACs) are important players in the cellular response to osmotic swelling[1–3]. These anion-selective channels are closed under resting conditions but become activated upon an increase of the cell volume resulting from the influx of water under hypotonic conditions[4,5]. Although the function of VRACs has been investigated for decades[6,7], their molecular identity was discovered only seven years ago, when members of the LRRC8 family were assigned as essential constituents of the channel[8–10]. All family members share a high mutual conservation with pairwise sequence identities exceeding 50% between the five human paralogs and 99% between human and murine LRRC8A orthologs[11]. In a cellular context, VRACs form heteromeric complexes all containing the obligatory LRRC8A subunit[8,9]. Although the exact composition of LRRC8 heteromers is currently unknown, they are believed to constitute a diverse family of ion channels whose subunit stoichiometry determines permeation and activation properties[12–14]. While the range of permeable substrates of channels containing the C-subunit is restricted to small anions, channels containing the D- or E- subunits also conduct larger molecules including osmolytes, amino acids, and anti-cancer drugs[12,15–17].

The general architecture of LRRC8 channels was revealed from homomeric structures of the LRRC8A[18–21] and LRRC8D[22] subunits. Although such homomeric assemblies are not observed in a cellular context, the subunits form hexamers and, in case of LRRC8A, they function as ion channels with compromised activation properties[18,23,24]. With respect to their structure, homomeric LRRC8A channels appear to exhibit general features that are also observed in heteromeric proteins[18]. This assumption is based on a low-resolution structure obtained from a preparation of LRRC8 oligomers containing A and C subunits. It refers to the hexameric organization of channels and their general structural features whereas differences in the molecular details and distinct conformational properties are expected to persist between homo- and heteromers. LRRC8 channels share a modular organization consisting of a membrane-inserted pore domain and cytoplasmic leucine-rich repeat (LRR) domains[18]. In the pore domain, the ion conduction path running along the symmetry axis is constricted by a narrow extracellular region that resembles a selectivity filter, followed by pore-widening within the membrane. In contrast to the well-defined pore domain, the cytoplasmic LRR units are mobile as they are usually less well- resolved in cryo-EM reconstructions[18–21]. In spite of their poor definition and lack of symmetry in the majority of imaged particles, a considerable population showed a channel arrangement in which the C6 symmetry of the pore reduces to a C3 relationship within the LRR domains[18,19]. Such architecture demands adjacent domains to change their respective orientation in order to maximize mutual contacts. Next to the tight interface between pairs, a second, loose interface is created at alternating positions in the hexameric channels, which is characterized by large fenestrations between interacting LRR domains. The functional relationship between the C3-symmetric channel structure and conformations with asymmetric LRR domain arrangement is still unknown. Despite the wealth of structural information, the role of the cytoplasmic LRR domains for channel function has remained elusive. This lack of knowledge is concomitant with our poor understanding of mechanisms of how VRACs sense changes in their environment and how these are converted into conformational transitions leading to channel activation[10,25–29].

In the present study, we are interested in the relevance of the LRR domains for the regulation of VRACs and thus investigate the effect of their interaction with proteinaceous binders on channel activity. To this end, we select synthetic nanobodies (termed sybodies)[30] targeting the LRR domain of the obligatory LRRC8A subunit and identify proteins that either inhibit or enhance channel activity. The structural characterization of their complexes reveals the interaction-epitopes and distinct conformations in the homomeric LRRC8A channel induced by sybody binding. Together our results provide a set of specific modulators of LRRC8 channels and they emphasize the importance of the LRR domains as regulatory units that modulate channel activity by allosteric mechanisms.

## Results

**Selection of sybodies targeting different regions of LRRC8A.** In light of the ambiguity concerning the molecular organization of LRRC8 channels and their mechanism of activation, we were interested in the characterization of these currently unknown molecular properties. We thus attempted to generate protein-based binders that specifically target different regions of the channel. For the identification of interaction partners of homomeric murine LRRC8A, we engaged in the in vitro selection of sybodies[30,31]. This selection process allowed the identification of numerous unique binders from three libraries containing either a short (concave library), intermediate (loop library), or long (convex library) randomized complementary-determining region 3.

From the pool of sybodies enriched against the full-length LRRC8A channel, we subsequently focused on a subset of binders that target its cytoplasmic domain. For this purpose, we have expressed and purified the soluble LRR domain of LRRC8A, which is monomeric and thus does not exhibit any of the inter-subunit interactions observed in hexameric channels. Using an ELISA setup with the soluble LRR domain as target, we identified two unique sybodies of the concave library (termed Sb1[LRRC8A] and Sb5[LRRC8A], short Sb1 and Sb5), two of the loop library (termed Sb2[LRRC8A] and Sb3[LRRC8A], short Sb2 and Sb3) and one of the convex library (termed Sb4[LRRC8A], short Sb4, Fig. 1a, Supplementary Fig. 1a). To further characterize the interaction between the five selected sybodies and LRR domains of different paralogs, we studied the elution behavior of complexes assembled from purified components on size-exclusion chromatography and analyzed peak fractions by SDS-PAGE. In this analysis, we detected the formation of stable complexes for all five sybodies with the LRR domain of LRRC8A but no interaction with the LRR domains of LRRC8C and D (Fig. 1b, Supplementary Fig. 1b). Finally, we quantified the interaction of the five sybodies to the isolated cytoplasmic LRR domains of the three paralogs by surface plasmon resonance spectroscopy (SPR). These investigations revealed that all five binders target the LRR domain of LRRC8A with dissociation constants in the nanomolar range with Sb1, Sb2, and Sb4 showing the highest, Sb5 an intermediate and Sb3 the lowest affinity (Fig. 1c–g, Table 1). Among the five sybodies, the interaction with Sb2 is distinguished by the slowest dissociation rate $k_{off}$ (with a residence time of over 300 s in the dissociation phase, Fig. 1c, Table 1). As for gel-filtration experiments, no binding to the LRR domains of LRRC8C and LRRC8D was detected in the investigated concentration range (Supplementary Fig. 1c), which further emphasizes the high specificity of the interaction for the LRRC8A subunit.

**Characterization of the modulatory properties of LRRC8A domain binders.** After the identification of sybodies that target the cytoplasmic LRR domain of VRACs, we were interested whether any of these binders would affect the functional properties of the channel. To this end, we have studied the activation of endogenous VRAC currents in HEK293 cells by patch-clamp electrophysiology in the whole-cell configuration. HEK293 cells show a strong current response mediated by heteromeric

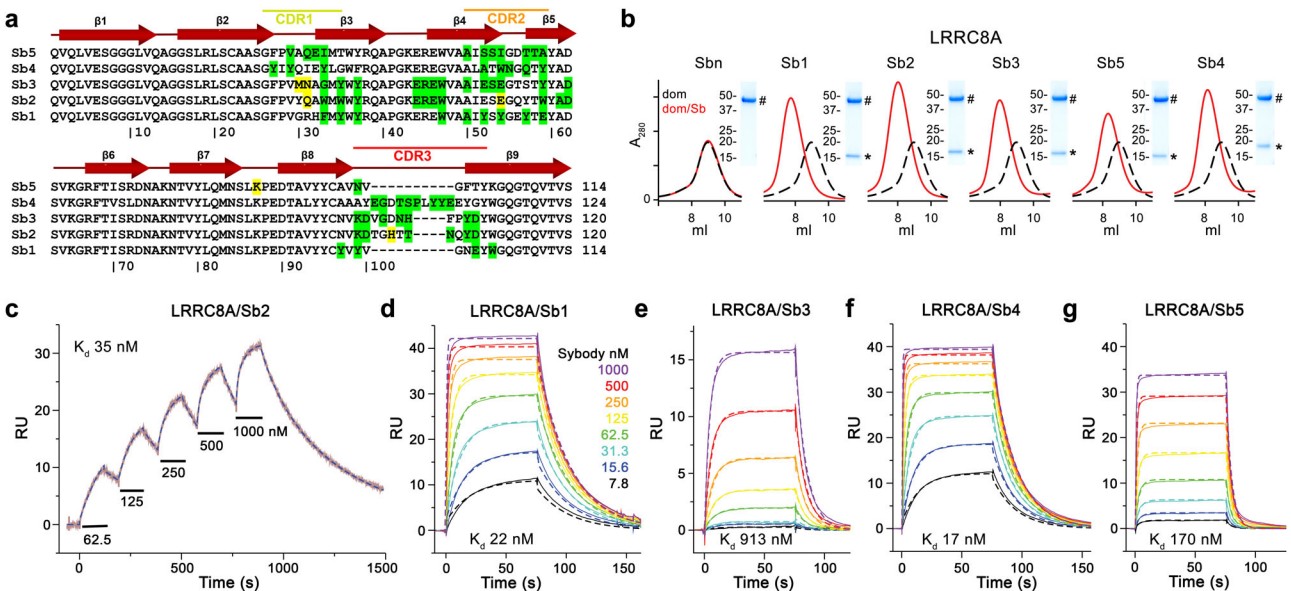

**Fig. 1 Biochemical characterization of sybodies targeting the LRR domains. a** Sequence alignment of the five modulatory sybodies. Residues involved in the interaction with the primary subunit are highlighted in green, contacts to the secondary subunit in yellow. Secondary structure elements and location of CDRs are indicated above. **b** Sections of size-exclusion chromatograms showing the elution of the LRR domain of LRRC8A (black dashed line) and its complex with interacting sybodies (red). Sbn refers to the mixture of the LRRC8A domain with a sybody targeting a protein unrelated to the LRRC8 family. Insets show SDS-PAGE gels of the peak fractions with the LRR domain of LRRC8A (#) and co-eluted sybodies (*) indicated. Numbers refer to the molecular weight (kDa). **c–g** SPR experiments of sybodies binding to immobilized LRR domains of LRRC8A. **c** Characterization of the binding properties of sybody Sb2. Bars indicate application of the sybody at increasing concentrations (nM). **d** Affinity determination of sybodies Sb1, **e** Sb3, **f** Sb4, and **g** Sb5. **d–g** Individual traces show association and dissociation of sybodies at concentrations indicated in **d** and coded in the same color. **c–g** Dashed lines superimposed on the respective recordings and dissociation constants were obtained from a fit to a 1:1 binding model.

**Table 1 Kinetic and dissociation constants of LRRC8A-LRR domain-sybody interactions obtained by SPR.**

|   |   | $K_D$ (nM) | $k_{on}$ (mol$^{-1}$ s$^{-1}$) | $k_{off}$ (s$^{-1}$) |
|---|---|---|---|---|
| **Sb1** | 1 | 24 | 1.25E+06 | 0.030 |
|  | 2 | 22 | 1.66E+06 | 0.037 |
|  | **ave.** | **23** | **1.5E+06** | **0.034** |
| **Sb2** | 1 | 35 | 8.28E+04 | 0.003 |
|  | 2 | 60 | 4.99E+04 | 0.003 |
|  | **ave.** | **48** | **6.6E+04** | **0.003** |
| **Sb3** | 1 | 800 | 1.30E+05 | 0.104 |
|  | 2 | 913 | 1.23E+05 | 0.113 |
|  | **ave.** | **857** | **1.3E+05** | **0.109** |
| **Sb4** | 1 | 27 | ND | ND |
|  | 2 | 17 | 1.72E+06 | 0.064 |
|  | **ave.** | **22** | **-** | **-** |
| **Sb5** | 1 | 175 | 1.14E+06 | 0.199 |
|  | 2 | 170 | 1.43E+06 | 0.242 |
|  | **ave.** | **173** | **1.3E+06** | **0.221** |

Data were fitted to a single binding site model. Table displays results of two independent biological replicates. Their average is shown in bold (**ave.**).
The bold values display the average of the two biological replicates.

channels of the LRRC8 family upon either cell swelling or the reduction of the intracellular ionic strength, although the relationship between both activation modes and the requirement of a certain degree of swelling as a prerequisite for channel opening has remained controversial[18,23,32,33]. We have previously used a protocol that relies on a reduced intracellular ionic strength in osmotically balanced conditions in combination with high ATP and low divalent ion concentrations, which synergistically leads to robust channel activation[18,29], and employed this protocol in the present study (Fig. 2, Supplementary Figs. 2 and 3). Evoked

currents are strongly anion-selective, slightly outwardly-rectifying and show a pronounced voltage-dependent inactivation at positive potentials[8,9,18] (Supplementary Figs. 2d, f and 3a, b). To characterize the modulation of VRAC activity by Sb1, we have added the sybody to the solution of the patch-clamp electrode to permit its diffusion into the cytoplasm after establishment of the whole-cell configuration. Following the activation of VRAC currents in response to a decreased (125 mM) intracellular salt concentration, we consistently observed an about fourfold reduction of current density compared to controls, thus suggesting that Sb1 might act as inhibitor of the channel (Fig. 2a, Supplementary Fig. 3a, b). In a next step, we investigated whether the expression of sybodies in the cytoplasm of HEK293 cells would lead to a similar inhibition of endogenous VRAC currents. We thus transfected HEK293 cells with a construct of the binder containing a C-terminal fusion of Venus-YFP. Such fusion-proteins expressed in the cytoplasm of HEK cells, termed intra-bodies, were shown to fold and recognize their intracellular targets[34–36]. In case of Sb1, we observed expression of the construct as judged by the strong YFP fluorescence inside the cell. To exclude the possibility that the sybody expression has perturbed the localization of VRACs within the cell, we quantified the fraction of channels at the plasma membrane by surface-biotinylation and found very similar protein levels as for non-transfected cells (Supplementary Fig. 3c, d). In recordings measured under activating conditions, we did not observe any current response, irrespectively of whether activation proceeded by swelling using a previously described protocol[8] or exposure of the cytoplasm to 125 mM salt (in conjunction with high ATP and low $Ca^{2+}$ and $Mg^{2+}$ concentrations, Fig. 2a, b, Supplementary Fig. 2a–c). A small and concentration-dependent response was observed upon further reduction of the salt concentration to 100 and 75 mM (Fig. 2b). In contrast, unaltered currents compared to wild type (WT) were recorded upon transfection with a construct

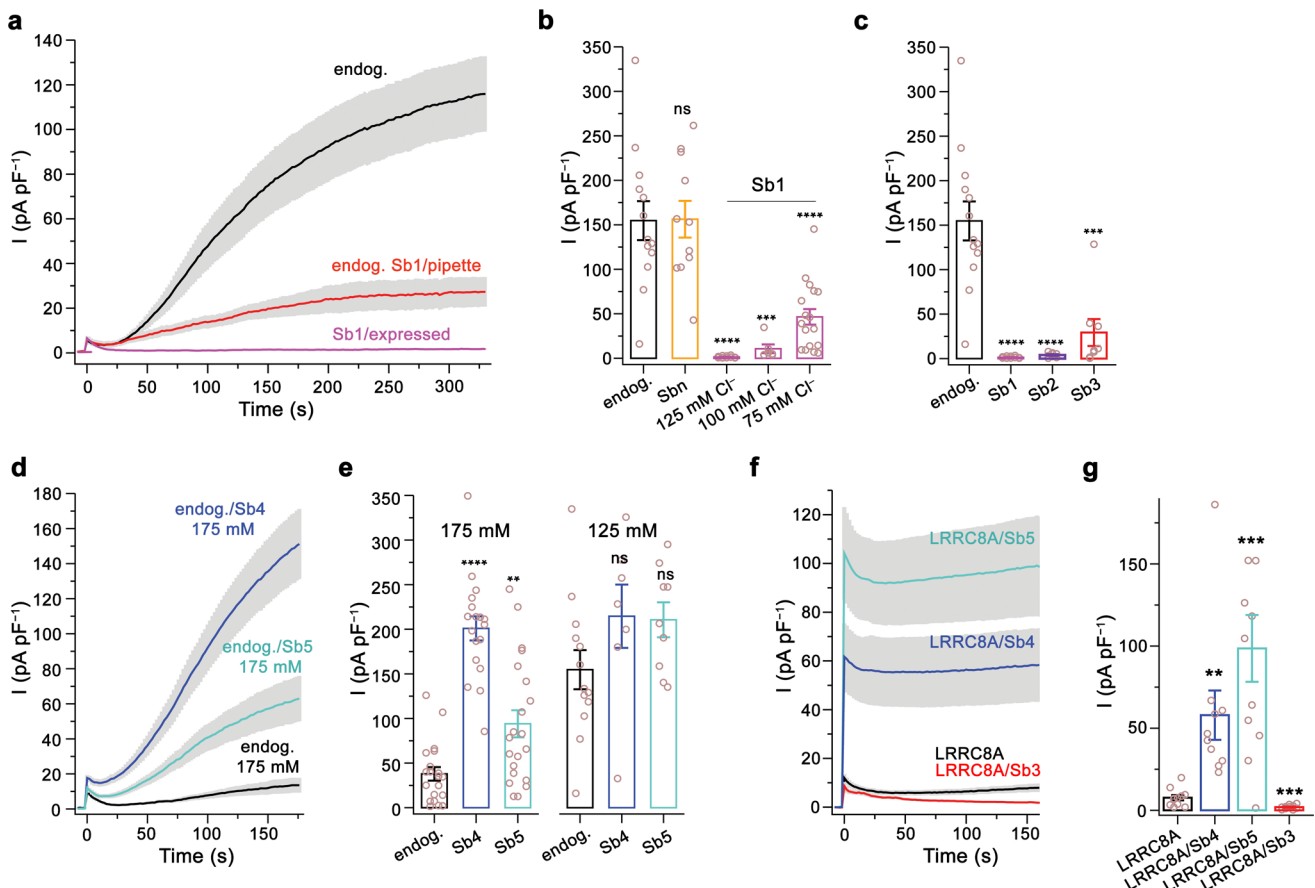

**Fig. 2 Functional characterization of sybodies targeting the LRR domains. a** Average current density of HEK293 cells in response to a decrease of the intracellular ionic strength. Currents from WT cells (endog.) with a pipette solution containing 125 mM salt (black, $n = 11$), additional 1 µM of Sb1 (red, $n = 10$) or from cells expressing Sb1 (magenta, $n = 12$). **b** Currents of cells expressing Sb1 at indicated intracellular ion concentrations (125 mM $Cl_i^-$, $n = 12$; 100 mM $Cl_i^-$, $n = 6$; 75 mM $Cl_i^-$, $n = 18$). Non-transfected cells (endog., $n = 13$) and cells expressing Sbn ($n = 11$), both recorded at 125 mM $Cl_i^-$, are shown for comparison. **c** Currents (at 125 mM $Cl_i^-$) from cells expressing different inhibitory sybodies (endog, $n = 13$; Sb1, $n = 12$; Sb2, $n = 8$; Sb3, $n = 8$). **d** Average response (at 175 mM $Cl_i^-$) of HEK293 cells expressing the potentiating sybodies Sb4 (blue, $n = 17$) and Sb5 (cyan, $n = 21$) compared to WT (black, $n = 20$). **e** Current-densities of cells expressing Sb4 and Sb5 in comparison to WT cells at 175 mM (endog., $n = 20$; Sb4, $n = 17$; Sb5, $n = 21$) and 125 mM $Cl_i^-$ (endog., $n = 13$; Sb4, $n = 7$; Sb5, $n = 9$). **f** Average response of $LRRC8^{-/-}$ cells expressing mouse LRRC8A (black, $n = 11$) or LRRC8A and the respective sybodies Sb4 (blue, $n = 10$), Sb5 (cyan, $n = 11$) and Sb3 (red, $n = 11$) at 75 mM $Cl_i^-$. **g** Mean current-densities of $LRRC8^{-/-}$ cells expressing mouse LRRC8A (black) and the indicated sybodies (LRRC8A, $n = 11$; Sb4, $n = 10$; Sb5, $n = 11$; Sb3, $n = 11$). Currents are recorded at 100 mV (**a**, **d**, **f**, **g**) or 80 mV (**b**, **c**, **e**) by patch-clamp electrophysiology in the whole-cell configuration. **b**, **c**, **e**, **g** Values of individual measurements are displayed as circles, mean current levels as bars. Asterisks indicate significant deviations from WT (endog.) in a two-sided one sample $t$-test, ns refers to non-significant differences. (**b** Sbn $p = 0.96$, Sb1 125 mM $p < 0.0001$, Sb1 100 mM $p = 0.0004$, Sb1 75 mM $p < 0.0001$, **c** Sb2 $p < 0.0001$, Sb3 $p = 0.0006$, **d** Sb4 175 mM $p < 0.0001$, Sb5 175 mM $p = 0.0023$, Sb4 125 mM $p = 0.147$, Sb5 125 mM $p = 0.087$, **g** Sb4 $p = 0.038$, Sb5 $p = 0.0006$, Sb3 $p = 0.0009$). **a**–**g** Data are from independent biological replicates, errors are s.e.m.

encoding a control sybody that was selected to target an unrelated protein (Sbn, Fig. 2b, Supplementary Fig. 2). Collectively, our results demonstrate that the presence of Sb1 in the cytoplasm of HEK cells prevents activation of VRAC channels by interacting with their LRR domains. To investigate whether other identified sybodies targeting the LRR domains would also inhibit the channel, we have expressed them in HEK cells and recorded currents in response to the reduction of the intracellular salt concentration to 125 mM with the same protocol. In these experiments we found a pronounced inhibitory effect of the sybodies Sb2 and Sb3 but not of Sb4 and Sb5 (Fig. 2c–e). The apparent weaker inhibition by Sb3 is consistent with its lower binding affinity to the LRRC8A domain quantified in SPR experiments (Fig. 1e, Table 1). As for Sb1, the expression of neither of the other four sybodies has affected the targeting of the channels to the plasma membrane (Supplementary Fig. 3c, d). At 125 mM salt, the average current density in cells transfected with

sybodies Sb4 and Sb5 is increased by about 50% compared to WT (Fig. 2e), which suggests that both proteins could potentiate activity. To further investigate this property, we have recorded current at elevated (i.e. 175 mM) ionic strength where we expected the effect of potential activators to be enhanced. At such conditions, we found very low response in non-transfected cells and a pronounced, on average fivefold increase of currents in case of cells expressing Sb4 and an on average 2.3-fold increase upon expression of Sb5 (Fig. 2d, e, Supplementary Fig. 3e). Finally, we have investigated whether similar functional effects as observed for heteromeric VRACs would also be found for the homomeric channel LRRC8A, whose activation is weak and requires very low ion concentrations[18]. Upon coexpression of LRRC8A with the potentiating sybodies Sb4 and Sb5 in $LRRC8^{-/-}$ cells, we recorded large, instantaneous, and anion-selective currents that are several-times increased compared to cells expressing LRRC8A alone (Fig. 2f, g, Supplementary Fig. 3f, g). Conversely, the low currents

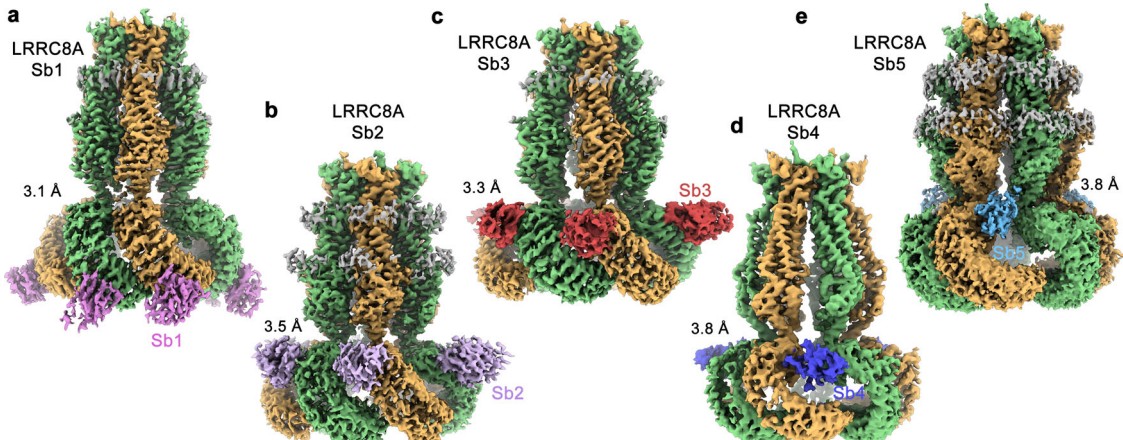

**Fig. 3 Cryo-EM density of LRRC8A-sybody complexes. a** LRRC8A/Sb1 complex at 3.1 Å. **b** LRRC8A/Sb2 complex at 3.5 Å. **c** LRRC8A/Sb3 complex at 3.3 Å. **d** LRRC8A/Sb4 complex at 3.8 Å. **e** LRRC8A/Sb3 complex at 3.8 Å. **a–e** Subunits at left and right positions in the C3-symmetric channels are colored in green and beige, respectively. Sybodies are shown in unique colors, residual density of the detergent micelle in gray. Binding of sybodies to all six subunits in case of the complexes with Sb1, Sb2 and Sb3, and to the subunits at r-positions in case of complexes with Sb4 and Sb5 is apparent.

of LRRC8A were further attenuated upon coexpression with the inhibitory sybody Sb3 (Fig. 2f, g, Supplementary Fig. 3f, g). Together, our data provide strong evidence that Sb1, Sb2, and Sb3 act as allosteric VRAC inhibitors and Sb4 and Sb5 as allosteric activators and that the distinct functional phenotypes of sybodies observed for LRRC8 heteromers are extended towards homomeric LRRC8A channels.

**Structural basis for the interaction with the inhibitory sybody Sb1**. To gain further insight into the sybody-VRAC interactions, we determined the structures of their complexes with homomeric channels composed of the obligatory subunit LRRC8A by cryo-electron microscopy (cryo-EM, Fig. 3, Supplementary Figs. 4–10, Table 2). Due to the high subunit selectivity of all five binders and their ability to target the isolated LRRC8A domain (Fig. 1, Supplementary Fig. 1), we expect these structures to also depict the binding mode of sybodies to the A-subunits in heteromeric VRAC channels.

First, we were interested in the interaction of a VRAC channel with an inhibitory sybody and hence determined the structure of the LRRC8A/Sb1 complex. The data is of high quality and allowed reconstruction of a map that extends to 3.1 Å for the entire complex and 2.7 Å for the pore domain (Supplementary Fig. 4). A large population of the particles (i.e. 26% of the particles used for 3D classification) shows a similar C3-symmetric structural arrangement as previously observed for the apo protein (Fig. 4a–c, Supplementary Fig. 4d, e). Other classes (in total encompassing 74% of the classified particles) show a well-defined pore domain but different degree of mobility of the cytoplasmic LRR domains). In the C3-symmetric structure, the densities of sybodies define the interaction of the binder with the channel at the lower part of the cytoplasmic domain towards the intracellular side (Fig. 3a, Supplementary Fig. 10a, b). In contrast to the apo protein, where the LRR domains were mobile and thus poorly defined in the cryo-EM density of the threefold symmetric channel conformation, in the LRRC8A/Sb1 complex these domains and their interacting sybodies are much better resolved (Supplementary Fig. 4d–h). The focused refinement on a symmetry-expanded dataset of a pair of interacting domains with bound sybodies yielded cryo-EM density at 2.8 Å, which allowed a detailed characterization of the complex (Supplementary Figs. 4h and 10a, b). In this substructure, the sybodies bind to the convex outside of the horseshoe-shaped domain (Fig. 4a–c). They target an epitope located on repeats 8–11 and bury 1420 Å²

of the combined molecular surface (Fig. 4d, Supplementary Fig. 11a). As intended by the design of the concave sybody library, the interface encompasses residues from β-strands 3, 4, 5 and 8 on the flat face of the binder involving residues from all three CDRs (Figs 1a and 4e). As the epitopes on the two LRR domains are separated from each other, sybodies interact in the same manner with either domain without contacts between neighboring binders (Fig. 4a–c). On the LRR domain, the residues buried in the interface are predominantly hydrophilic, whereas on the sybody they are dominated by aromatic sidechains (Fig. 4e, f). The high-resolution map of the domain pair also defines the conformation of residues that are buried in the interface between the two LRR domains, which were not resolved in the cryo-EM reconstruction of the apo protein (Supplementary Figs. 10b and 11b–e).

In the C3-symmetric LRRC8A structure, tightly interacting LRR domain pairs are denoted as left (l) and right (r) subunits according to their relative position when viewed from the outside of the channel[18] (Fig. 4a, b). Their respective orientations differ by a 42° rotation around a hinge located at the boundary to the pore, which maximizes their mutual interaction and buries 1507 Å² of the combined molecular surface (Supplementary Fig. 11b). Since several of the buried residues are charged, the interaction might be dominated by electrostatic contributions (Supplementary Fig. 11c–e). In addition, the LRRC8A/Sb1 complex also reveals the conformation of the C-terminal part of a mobile loop connecting the cytosolic helices CLH1 and CLH2 on the r-subunit, which was poorly defined in the apo structure (Fig. 4b, Supplementary Figs 10b and 11f). Since this loop carries phosphorylation sites and was suggested to play a role in channel regulation[10,25], the observed interaction could be of functional significance.

**Structural basis for the interaction with the inhibitory sybodies Sb2 and Sb3**. After identifying the binding mode of Sb1, we were interested in the structural properties of other inhibitory sybodies and thus studied the interaction of the channel with Sb2 and Sb3. The structures of the LRRC8A/Sb2 and LRRC8A/Sb3 complexes are both of high quality and define the mutual relationship between the channels and their interaction partners (Fig. 3b, c, Supplementary Figs. 5 and 6). Like in case of Sb1, the interaction of Sb3 with LRRC8A has stabilized the cytoplasmic LRR domains and thus allowed a detailed structural characterization of its binding mode (Supplementary Figs. 6d–h and 10c, d). In contrast,

**Table 2 Cryo-EM data collection, refinement, and validation statistics.**

| | Dataset 1 LRRC8A/Sb1 (EMD-13202) (PDB 7P5V) | Dataset 2 LRRC8A/Sb2 (EMD-13203) (PDB 7P5W) | Dataset 3 LRRC8A/Sb3 (EMD-13208) (PDB 7P5Y) | Dataset 4 LRRC8A/Sb4 (EMD-13212) | Dataset 5 LRRC8A/Sb4$_{0.5}$ (EMD-13213) (PDB 7P60) | Dataset 6 LRRC8A/Sb5 (EMD-13230) (PDB 7P6K) |
|---|---|---|---|---|---|---|
| **Data collection and processing** | | | | | | |
| Microscope | FEI Titan Krios | FEI Titan Krios | FEI Titan Krios | FEI Titan Krios | FEI Titan Krios | FEI Titan Krios |
| Camera | Gatan K3 GIF | Gatan K3 GIF | Gatan K3 GIF | Gatan K3 GIF | Gatan K3 GIF | Gatan K3 GIF |
| Magnification | 130,000 | 130,000 | 130,000 | 130,000 | 130,000 | 130,000 |
| Voltage (kV) | 300 | 300 | 300 | 300 | 300 | 300 |
| Electron exposure (e⁻/Å²) | 61 | 61 | 61 | 61 | 61 | 61 |
| Defocus range (μm) | −2.4 to −1.0 | −2.4 to −1.0 | −2.4 to −1.0 | −2.4 to −1.0 | −2.4 to −1.0 | −2.4 to −1.0 |
| Pixel size (Å)* | 0.651 (0.326) | 0.651 (0.326) | 0.651 (0.326) | 0.651 (0.326) | 0.651 (0.326) | 0.651 (0.326) |
| Initial number of micrographs (no.) | 5494 | 5633 | 6475 | 6416 | 4869 | 5199 |
| Initial particle images (no.) | 579,709 | 330,072 | 756,017 | 313,757 | 507,983 | 530,412 |
| Final particle images (no.) | 59,962 | 65,959 | 76,350 | 14,718 | 38,121 | 48,917 |
| Symmetry imposed | C3 | C3 | C3 | C3 | C3 | C3 |
| **Map resolution FL, TM (Å)** | | | | | | |
| FSC threshold 0.143 | 3.1, 2.7 | 3.5, 3.0 | 3.3, 2.9 | 7.7, - | 3.8, 3.5 | 3.8, 3.5 |
| Map resolution range (Å) | 2.6–6 | 2.8–12 | 2.9–6 | 7–10 | 3.1–8 | 3.3–8 |
| **Refinement** | | | | | | |
| **Model resolution (Å)** | | | | | | |
| FSC threshold 0.5 | 3.26 | 3.8 | 3.5 | - | 3.9 | 5.7 |
| Map sharpening b-factor (Å²) | −36 | −89 | −69 | | −76 | −66 |
| **Model composition** | | | | | | |
| Non-hydrogen atoms | 41,280 | 41,256 | 41,106 | | 38,409 | 38,109 |
| Protein residues | 5034 | 5028 | 5028 | | 4680 | 4650 |
| **B factors (Å²)** | | | | | | |
| Protein | 47.0 | 138.1 | 34.9 | | 138.6 | 76.8 |
| **R.m.s. deviations** | | | | | | |
| Bond lengths (Å) | 0.004 | 0.002 | 0.002 | | 0.004 | 0.002 |
| Bond angles (°) | 0.514 | 0.487 | 0.491 | | 0.552 | 0.469 |
| **Validation** | | | | | | |
| MolProbity score | 2.1 | 2.3 | 2.1 | | 2.5 | 2.2 |
| Clashscore | 9.0 | 10.7 | 10.3 | | 14.6 | 9.5 |
| Poor rotamers (%) | 3.5 | 3.7 | 2.6 | | 5.5 | 3.9 |
| **Ramachandran plot** | | | | | | |
| Favored (%) | 96.6 | 95.6 | 96.5 | | 96.0 | 96.4 |
| Allowed (%) | 3.4 | 4.4 | 3.5 | | 4.0 | 3.6 |
| Disallowed (%) | 0.0 | 0.0 | 0.0 | | 0.0 | 0.0 |

*Values in parentheses indicate the pixel size in super-resolution.

the same units show increased mobility in the LRRC8A/Sb2 complex as reflected in the lower local resolution of the cryo-EM density of the LRR domains and their attached sybodies, which are both weaker and less well defined than the transmembrane part (Supplementary Figs. 5d–f and 10e, f). Consequently, we had to rely on a homology model of Sb2 to describe its interactions with the channel, which should thus be considered as tentative.

Irrespective of differences in their sequence and their effect on the channel structure, both sybodies recognize the same epitope on the LRR domain (Fig. 5a–e, Supplementary Fig. 11a). Similar to Sb1, Sb2, and Sb3 both target individual LRRC8A subunits on the convex side of the LRR domain but, in contrast to former, they bind to a region that is located closer to the transmembrane domain (Fig. 5a–d). For complexes with Sb2 and Sb3, the interaction involves residues on repeats 3–6 on the LRR domain (Fig. 5e). In the well-defined interactions of the LRRC8A/Sb3 complex, 1495 Å² of the combined molecular surface are buried by the sybody targeting the l-subunit and 1866 Å² by the sybody

targeting the r-subunit. The latter interface is increased in both Sb2 and Sb3 complexes due to contacts of the bound sybody with the neighboring l-domain (Fig. 5b, d). Remarkably, the observed binding interferes with the interaction of the CLH1-CLH2 loop, whose C-terminal end was observed to attach to the LRR domain in the LRRC8A/Sb1 complex (Fig. 4b, Supplementary Fig. 11f), with potential functional consequences. Despite their distinct location, the general interaction modes of all three inhibitory sybodies share resemblance in that they make contacts via their flat surface involving both conserved and variable residues residing on β-strands and adjacent loops of all three CDRs (Figs 1a, 4d, e and 5e, f, h). As opposed to Sb1, the binding interfaces of Sb2 and Sb3 contain fewer aromatic amino acids (i.e. 5 and 6 aromatic residues in Sb3 and Sb2 respectively, compared to 10 aromatic residues in Sb1) and are generally more hydrophilic (Fig. 5f, h). In light of the similar binding mode of Sb2 and Sb3, the large difference in their dissociation constants, which is primarily a consequence of the 20-times faster off-rate of

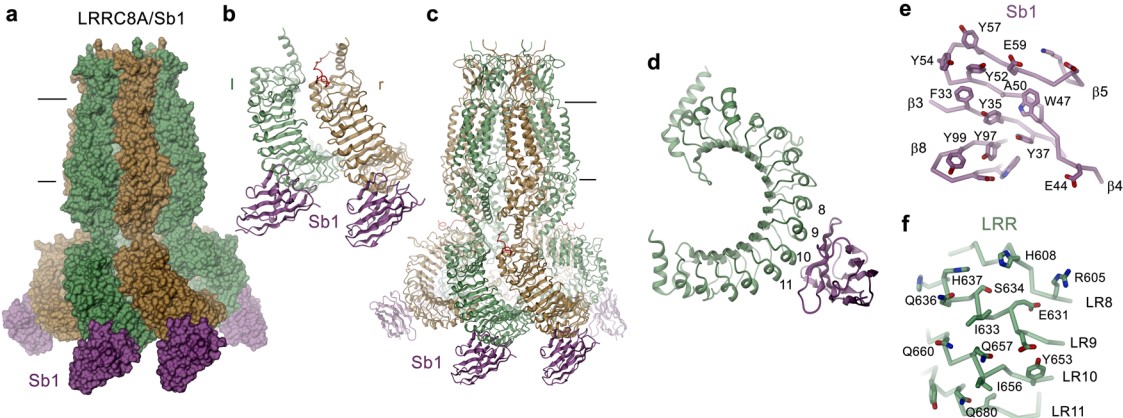

**Fig. 4 Structure of LRRC8A in complex with the inhibitory sybody Sb1. a** Surface representation of the LRRC8A/Sb1 complex structure. **b** Structure of the dimer of interacting domains at the tight interface with bound sybody Sb1. Left (l) and right (r) positions are indicated. **c** Ribbon representation of the LRRC8A/Sb1 complex. **a**, **c** The view is from within the membrane with membrane boundaries indicated. **d** Ribbon representation of a single LRR domain with sybody Sb1 bound. Repeats contacted by Sb1 are labeled. **e** View on the interaction interface of Sb1 and **f** the LRRC8A domain. The protein is shown as Cα trace with the sidechains of interacting residues displayed as sticks.

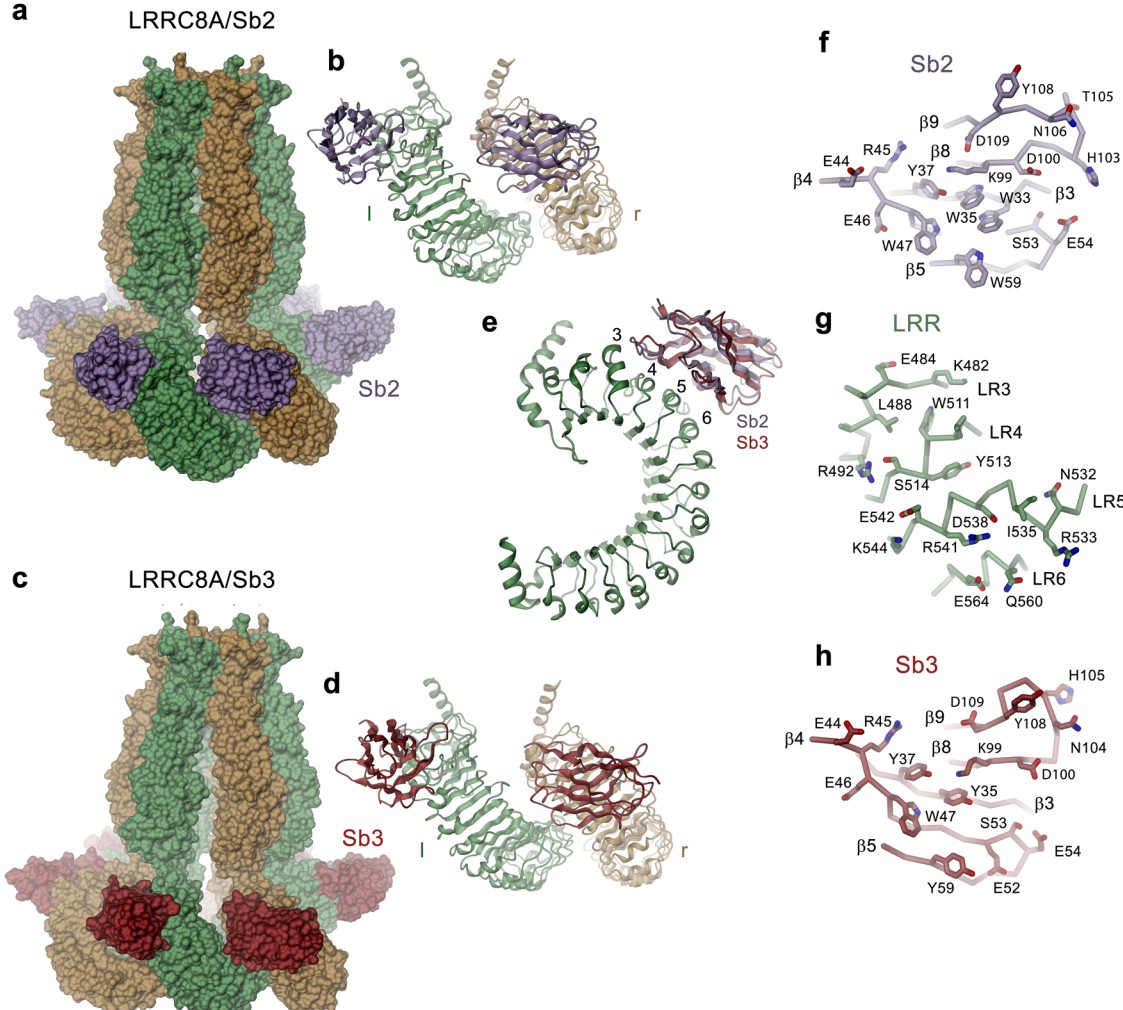

**Fig. 5 Structure of LRRC8A in complex with the inhibitory sybodies Sb2 and Sb3. a** Surface representation of the LRRC8A/Sb2 complex structure. **b** Structure of the dimer of interacting domains at the tight interface with bound sybody Sb2. **c** Surface representation of the LRRC8A/Sb3 complex structure. **d** Structure of the dimer of interacting domains at the tight interface with bound sybody Sb3. **b**, **d** Left (l) and right (r) positions are indicated. **e** Ribbon representation of a single LRR domain with bound sybodies Sb2 and Sb3. Repeats contacted by both sybodies are labeled. **f** View on the interaction interface of a homology model of Sb2 based on the Sb3 structure, **g** the LRRC8A domain and, **h** Sb3. The protein is shown as Cα trace with the sidechains of interacting residues displayed as sticks.

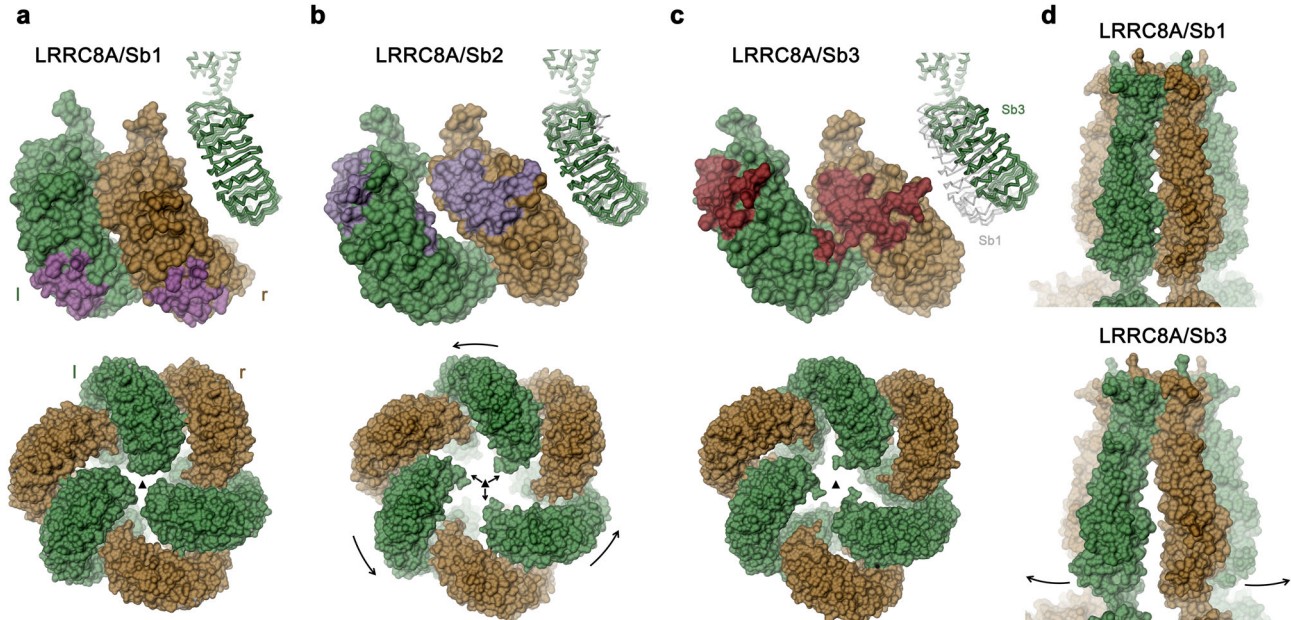

**Fig. 6 Conformational changes of LRRC8A in response to inhibitory sybody binding. a** LRRC8A/Sb1 complex, **b** LRRC8A/Sb2 complex, **c** LRRC8A/Sb3 complex. **a**–**c** Shown are surface representations of domain pairs at the tight interface (top) with contact regions with bound sybodies colored and the hexameric assembly of the domains viewed from the cytoplasm with symmetry axis indicated as triangle (bottom). Insets (top right) display Cα trace of the right subunit of the respective structure, which is superimposed in **b** and **c** on the respective structure of the complex with sybody Sb1. **d** Surface representation of the pore domain of the LRRC8A/Sb1 (top) and the LRRC8A/Sb3 complex (bottom) viewed from within the membrane.

Sb3 compared to Sb2 (Fig. 1c, e, Table 1) is noteworthy and is presumably caused by single specific interactions of both binders.

Despite the common inhibitory phenotype of the sybodies Sb1, Sb2, and Sb3, the channels in their respective C3-symmetric complex structures adopt distinct conformations that can be approximated as rigid body movements of pore and LRR domains (Fig. 6, Supplementary Video 1). In Sb2 and Sb3 complexes, these conformational changes are the consequence of the relaxation of a steric clash of the respective sybodies bound to the r-subunit with the LRR domain of the neighboring l-subunit, which would prevail in the conformation observed for the apo protein and the Sb1 complex (Fig. 5a, c and 6b, c).

In the LRRC8A/Sb2 complex, the conformational differences compared to the apo protein can be approximated by a small (7°) rotation of the LRR domain in the r- and a larger (17°) rotation of the LRR domain in the l-position around a hinge located at the respective connection to the pore domain (Fig. 6, Supplementary Video 1). The described movement causes the dissociation of contacts at the tight interface between two domains, which leads to a reduction of the buried surface area from 1390 to 447 Å² (Fig. 6a, b). The consequent opening of a gap in the center of interacting subunits leaves only few contacts at the N- and C-terminal ends of the respective LRR domains (Supplementary Fig. 11g). Due to the described movements, the three l-domains, which approach each other at the threefold axis of symmetry in the apo protein, have retracted by 16 Å to open a gap at the intracellular side (Fig. 6b, Supplementary Video 1).

In the LRRC8A/Sb3 complex, the structural rearrangements of the LRR domains are even more pronounced and they are also accompanied by changes of the pore. Whereas the conformation of the r-subunit resembles the LRRC8A/Sb2 complex, the enhanced (26°) reorientation of the LRR domain of the l-subunit results in its movement towards the membrane (Fig. 6c, Supplementary Video 1). The concurrent redistribution of interactions establishes contacts between repeats 10–16 of the

l-domain and repeats 1–8 at the contacted r-domain, which bury 904 Å² of the combined molecular surface (Supplementary Fig. 11h, i). The described relocation of the LRR domains appears to be coupled to the pore where the interface between neighboring l- and r-domains is disrupted at the intracellular side and both domains are splayed open by an 8° rotation around an axis that is located close to the extracellular sub-domain, which remains unchanged in both structures (Fig. 6d, Supplementary Video 1). The resulting dissociation of contacts results in a pronounced gap at the intracellular side (Fig. 6d). The described conformational changes only affect the relationship between l- and r-subunits at the tight interface, whereas the interactions at the loose interface remain unchanged. As a result, the C6 symmetry of the pore domain observed in the apo structure is reduced to a C3 relationship relating pairs of TM domains that have retained their mutual interactions.

Thus, although Sb1, Sb2, and Sb3 share a similar inhibitory phenotype and a related mode of interaction by binding to single domains at the convex face of the LRR domain, the common epitope of Sb2 and Sb3 and the corresponding site of Sb1 are located on opposite ends of the domain. Moreover, in both the Sb2 and Sb3 complex structures, the steric clashes between interacting LRRC8A subunits and bound sybodies stabilize distinct protein conformations in the homomeric channel with currently unclear relationship to its functional state.

**Structural basis for the interaction with the potentiating sybodies Sb4 and Sb5.** To explore the relationship between inhibitory sybodies and binders that potentiate channel activity, we have determined structures of LRRC8A in complex with the sybodies Sb4 and Sb5. Compared to complexes with Sb1, Sb2, and Sb3, these structures show divergent features with respect to the recognized epitopes and the observed binding stoichiometry. In contrast to the three inhibitory sybodies, which target each subunit of the hexameric LRRC8A channel by binding to the convex

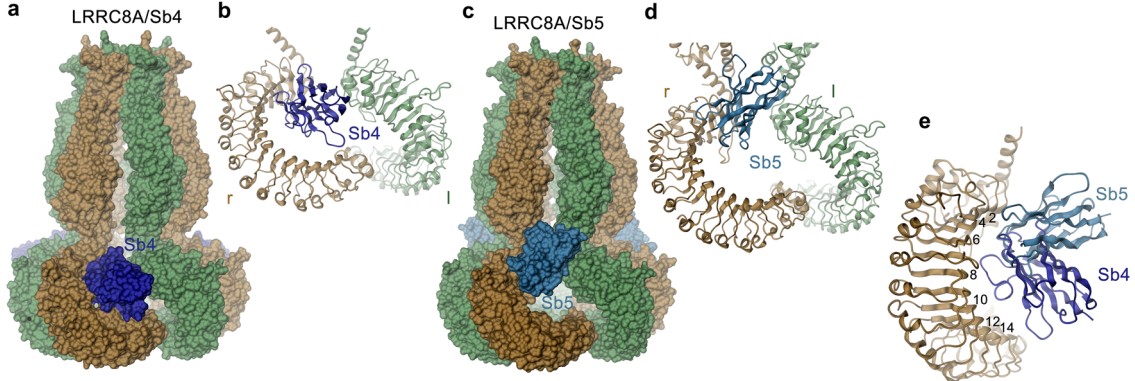

**Fig. 7 Structure of LRRC8A in complex with the potentiating sybodies Sb4 and Sb5. a** Surface representation of the LRRC8A/Sb4 complex structure. **b** Structure of the dimer of interacting domains at the loose interface with bound Sb4. **c** Surface representation of the LRRC8A/Sb5 complex structure. **d** Structure of the dimer of interacting domains at the loose interface with bound Sb5. **b, d** Left (l) and right (r) positions are indicated. **e** Ribbon representation of a single LRR domain with bound sybodies Sb4 and Sb5. Repeats contacted by one of the two sybodies are labeled. **a–e** Due to the low resolution of cryo-EM densities of binders in the respective maps, Sb4 and Sb5 structures are based on homology models of both sybodies.

face of the LRR domain, thereby stabilizing distinct conformations, the complexes with Sb4 and Sb5 display a high conformational heterogeneity where the LRR domains are poorly resolved in a majority of 3D classes (Supplementary Figs. 7 and 8). Besides the structures with high domain mobility, in both complexes we also observed C3-symmetric channel conformations, which display interactions that differ from inhibitory sybody complexes (Fig. 3d, e, Supplementary Figs. 7, 8). Whereas in these maps the pore domains are of high quality and the less well-defined LRR domains reveal the location of sybodies and their general interaction with channel epitopes, the resolution of the maps is insufficient for a detailed structural interpretation of sybody interactions (Supplementary Figs. 7g, h, 8e, f and 10g–j). The described interactions, which are based on homology models of the sybodies Sb4 and Sb5, are thus approximate.

In both, LRRC8A/Sb4 and LRRC8A/Sb5 complexes, the sybodies bind to alternating subunits as their epitopes are only accessible in LRR domains located at the r-position of tightly interacting domain pairs whereas they are hidden in the dimer interface in the l-subunit (Fig. 7). Their interaction with residues located on the edge between the flat face and the concave inside of the LRR domain involves repeats 2–14 in case of the Sb4 complex or 2–7 in case of the Sb5 complex (Fig. 7e, Supplementary Fig. 11a). The smaller interface and lack of aromatic residues contributed by Sb5 is congruent with its lower affinity compared to Sb4 (Fig. 1e and Supplementary Fig. 1c). Both sybodies target overlapping sites, yet with different binding modes and they bridge the large fenestrations between adjacent domains at the loose interface by approaching the juxtaposed l-subunits (Fig. 7a–d). The resulting conformational changes to accommodate the binders lead to the retraction of the LRR domains from the threefold axis, although to a smaller extent than observed in the LRRC8A/Sb2 and Sb3 complexes (Fig. 6a–c, Supplementary Fig. 11j, k, Supplementary Video 2). As in the LRRC8A/Sb3 complex, the conformational rearrangement of the LRR domains is coupled to the pore domain leading to the opening of intracellular subunit contacts, akin to conformational changes observed for the Sb3 complex, though this time at the loose interface (Supplementary Fig. 11j, k, Supplementary Video 2). Together, the structures of complexes with potentiating sybodies reveal an interaction where the binders target an epitope that is buried in one subunit of interacting domain pairs. The accessibility of this site in the observed C3-symmetric channel conformation will thus depend on the relative position of LRRC8A subunits in heteromeric channels.

## Discussion

In our study, we have investigated the role of the cytoplasmic LRR domains of VRACs for the regulation of their activity. These protein components form modular units of LRRC8 channels that resemble ligand-binding domains. Although their potential role as interaction platforms is evident, since structurally related units in other proteins are known to bind diverse small- and macromolecules[37–39], no ligands are currently known for this particular family of ion channels[10]. To characterize the effect of interacting proteins on the functional properties of VRACs, we have selected nanobodies from synthetic libraries[30,31], which specifically bind the cytoplasmic domain of the LRRC8A subunit with nanomolar affinity (Fig. 1, Supplementary Fig. 1, Table 1). As shown by patch-clamp electrophysiology, these sybodies modulate the activity of endogenous VRACs in HEK293 cells (Fig. 2). Whereas three of the selected binders were found to inhibit activity, two others showed a potentiating effect, which emphasizes the importance of these domains as regulatory units of LRRC8 channels.

Insight into the structural basis of VRAC modulation was obtained from cryo-EM structures of LRRC8A-sybody complexes. Irrespective of the fact that LRRC8A homomers are not found under physiological conditions, such assemblies form functional anion channels, although with compromised activation properties[18,23,24]. Moreover, the observed strong potentiation of LRRC8A activity by the sybodies Sb4 and Sb5 and its inhibition by Sb3 further emphasize an equivalent modulatory role of binders also in the context of homomeric channels (Fig. 2f, g). The respective complex structures thus likely display general properties of sybody interactions that might also extend towards heteromeric channels. Since the local concentration of the targeted A-subunit is increased compared to their heteromeric equivalents, we assume observed structural features to be even enhanced in homomeric channels. However, due to the unknown disposition of subunits in LRRC8 heteromers, we also expect unique properties of sybody interactions in endogenous heteromeric channels, which will have to be explored in future studies. In the five complexes, the sybodies show discrete binding modes, with the corresponding channel structures providing insight into accessible conformations of the LRR domains and their potential coupling to the transmembrane pore. The three inhibitory sybodies (Sb1, Sb2, and Sb3) target all subunits of the hexamer and they interact with epitopes located on two separated positions on the convex side of the LRR domain (Fig. 8a). Sybody Sb1 binds towards the C-terminus and stabilizes the threefold symmetric

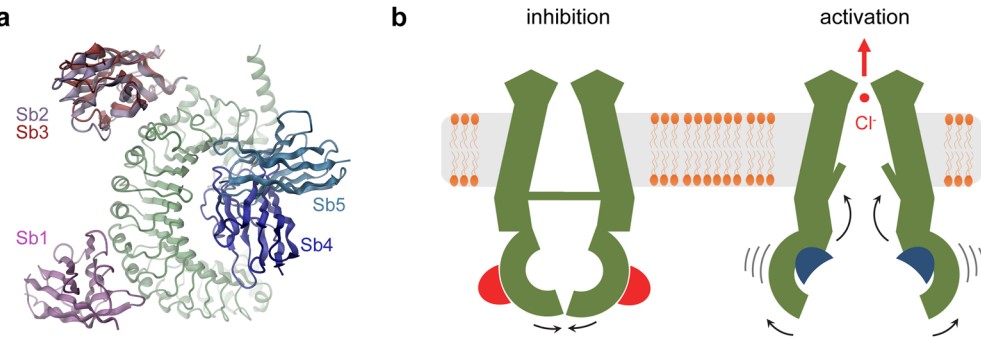

**Fig. 8 Potential mechanisms. a** Location of sybody binding sites on the LRR domain of LRRC8A. The proteins are shown as ribbon with different sybodies labeled. **b** Schematic depiction of a potential modulatory mechanism of channel function by sybody binding. Binding of inhibitory sybodies to the convex outside of the LRR domains (left) reduces flexibility of the domains, which stabilizes a closed channel conformation. Conversely, the binding of activating sybodies to the concave inside increases domain mobility, which is in some way transmitted to the pore region to open the ion conduction path.

channel conformation observed in the apo protein[18,19] (Fig. 4a–c). The improved density of this complex now defines the previously poorly resolved conformation of residues buried in the interface (Supplementary Fig. 10a, b). The large number of ionizable residues suggest a plausible dependence of interactions between LRR domains on the ionic strength that could be weakened by the shielding of charges at higher salt concentrations and a hypothetical interaction with divalent cations, although the role of such interactions will have to be clarified in future studies (Supplementary Fig. 11c–e). Conversely, sybodies Sb2 and Sb3 recognize a common epitope that is located closer to the membrane. Their binding induces conformational changes of the LRR domains, which in both cases lead to the dissociation of the tight domain interface and which in case of Sb3 extends towards the pore (Fig. 6b–d). Collectively, these structures display a set of conformations that could be adopted as a part of the regulatory mechanism. However, since all three sybodies share a similar inhibitory phenotype, the functional correspondence of the observed conformations remains unclear and it is unlikely that any of them represents an open channel.

In contrast to the stabilizing effect of the inhibitory sybodies Sb1 and Sb3, the two potentiating sybodies Sb4 and Sb5 appear to increase the overall mobility of the LRR domains, which is reflected in the poor resolution of the respective region in both complex structures (Supplementary Figs. 7, 8 and 10g–j). In C3-symmetric channel conformations, both sybodies bind at the loose interface to a site located at the edge between the flat face and the concave side of the domain (Figs. 7, 8a). In these structures, the sybodies solely target subunits on the r-position of interacting domain pairs as the epitope on the l-subunit is buried in the interface (Fig. 7a–d). The bridging to the juxtaposed domain pair causes a conformational rearrangement at the loose interface whereas the relationship between interacting LRR domains at the tight interface remains unaffected (Supplementary Fig. 11j, k). During activation, it is conceivable that the same sybodies would also target the epitope hidden in the LRR domain interface, thereby facilitating the dissociation of contacts and increasing domain mobility (Fig. 8b). The correlation between LRR domain mobility and channel activity is intriguing also in light of previous observations. C-terminal GFP fusion proteins, which might obstruct tight domain interactions were found to lead to an increased basal activity of the over-expressed channel[14]. Additionally, a study using complementary fluorescent proteins acting as FRET pairs fused to different LRRC8 subunits suggested a conformational change of the LRR domains during activation[28].

In case of the inhibitory sybody Sb3 and the potentiating sybodies Sb4 and Sb5, the rearrangements of the LRR domains

couple to the membrane-inserted part of the protein leading to the disruption of intracellular contacts and a breakdown of the C6 symmetry of the pore. The transition splays apart interacting subunits at the intracellular part and potentially might open the access of membrane lipids to the pore to modulate its conduction properties (Fig. 6d, Supplementary Fig. 11j, k, Supplementary Videos 1 and 2). Similar, yet less extensive features of the pore domain conformation were previously found in one of the structures obtained from LRRC8A in absence of binders[20] and a moderate symmetric expansion of the pore domain was observed for a population of the channel embedded in lipid nanodiscs[21]. Although not confined to a single functional phenotype of binders, these structures may illustrate a possible pathway for coupling from the LRR domain to the pore to modify a gate that impedes ion conduction in the closed conformation. The location of this gate has not yet been assigned with confidence but it might either involve the N-termini pointing towards the pore axis[19,27] or the narrow pore region located at the extracellular side[25,26] as mutations in both regions affect conduction and activation properties of the channel. The long-range nature of the observed transitions suggests that effects might potentially lead to changes in the conformation of the N-terminus and even extend towards the narrow extracellular filter, although its conformation appears unaltered in different structures obtained in this study.

In summary, our study has generated a diverse set of proteins that modulate VRACs by either inhibiting or potentiating their activity. The structures of their complexes have revealed the recognized epitopes and conformations of LRRC8A induced by sybody binding. While in a cellular context, the detailed interaction would depend on the currently unknown distribution of A subunits in LRRC8 heteromers, all structures display the intrinsic plasticity of the channel, which presumably underlies activation. However, in absence of a clear correlation with the modulatory phenotype of binders, the assignment of distinct conformations to functional states is at this stage ambiguous and their relevance in heteromeric channels still awaits investigation. In all cases, it is also possible that the sybodies act by preventing the binding of currently unknown interaction partners. Our data emphasize the importance of the cytoplasmic LRR domains of VRACs in modulating the activation of the pore domain by allosteric mechanisms. The generated set of interacting proteins will serve as important tools for future studies. These range from the structural characterization of heteromeric channels to their investigation in a cellular context and the development of potential therapeutic approaches aiming at the inhibition of VRAC channels in cerebral ischemia[40] and their activation in certain type of cancers to facilitate the uptake of drugs[41,42].

## Methods

**Expression constructs and cloning**. All constructs were generated using FX-cloning and FX-compatible vectors[43]. The constructs encompassing full-length murine LRRC8A and C, the LRRC8A-pore domain (PD) and the LRRC8A-cytosolic domain (LRR) were obtained from a previous study[18]. The LRR domain of LRRC8A encompassed residues 398-810 of mouse LRRC8A (NP_808393.1). The boundaries for the LRR domains of LRRC8C and LRRC8D were chosen analogously to LRRC8A with obtained constructs encompassing residues 396–803 and 442–858 of mouse LRRC8C (NP_598658.1) and LRRC8D (NP_001127951.1), respectively. For large-scale expression experiments, full-length LRRC8A and the LRRC8A-PD were cloned into a pcDX vector containing a C-terminal Rhinovirus 3C protease-cleavable linker followed by mCherry[44], a myc-tag and streptavidin-binding peptide[45] (SBP, pcDXc3ChMS). For patch-clamp experiments full-length LRRC8A and LRRC8C were cloned into an analogous vector not containing mCherry (pcDXc3MS). The LRR domains of LRRC8A, C and D were cloned into a pcDX vector containing an N-terminal SBP, a myc-tag followed by a Rhinovirus 3C protease-cleavable linker (pcDXn3MS). For periplasmic expression of sybodies in bacteria, sequences were cloned into pSBinit[30], an arabinose inducible vector harboring chloramphenicol resistance gene and containing an N-terminal pelB leader sequence and a C-terminal His₆-tag. For cytoplasmic expression in mammalian cells, sybodies were cloned into a pcDX vector containing a C-terminal Rhinovirus 3C protease-cleavable linker followed by Venus[46], a myc-tag and SBP (pcDXc3VMS).

**Protein expression and purification**. For cryo-EM analyses and binding tests, full-length LRRC8A, the LRRC8A-PD and LRR domains of LRRC8A, C and D were expressed in HEK293S GnTI⁻ cells[47] and purified by affinity chromatography on StrepTactin Superflow resin (IBA Lifesciences) and size-exclusion chromatography on a Superose 6 10/300 column (GE Healthcare)[18]. The wash- and size-exclusion chromatography buffers of the LRR domains of LRRC8A, C and D contained 10 mM HEPES pH 7.5, 150 mM NaCl, 0.5 mM EDTA and 0.25 mM n-Dodecyl-β-D-Maltoside (DDM, Anatrace) and tag cleavage with Rhinovirus 3C protease was performed in solution, similarly to full-length LRRC8A. For sybody selection and ELISA screening, full-length LRRC8A and the LRRC8A-PD were purified in presence of glycol-diosgenin (GDN, Anatrace) at a concentration of 2% for extraction and 100 µM for subsequent steps. The sybody expression constructs were transformed into *E. coli* MC1061 and bacteria were grown in TB medium (Sigma) supplemented with 35 µg ml⁻¹ chloramphenicol (Sigma) at 37 °C. At an OD₆₀₀ of 0.5, the temperature was decreased to 22 °C and after 30 min incubation, expression was induced by addition of L-arabinose (Sigma) to a final concentration of 0.02%. After 16–18 h, cultures were harvested by centrifugation and pellets were either used immediately or stored (−20 °C) until further use. Stored bacterial pellets were thawed on ice and resuspended in 100 mM Tris pH 8.0, 1 mM EDTA, 30% sucrose supplemented with 1 mM PMSF (10 ml of buffer per 1 g of pellet) and incubated at room temperature under gentle agitation for 30 min. The suspension was diluted with 4 volumes of 150 mM NaCl, 1 mM MgCl₂ and incubated for another 15 min. All subsequent purification steps were performed at 4 °C. The suspension was centrifuged for 30 min at 8000 g. The supernatant was supplemented with 15 mM imidazole and Ni-NTA resin (4 ml of 50% slurry for material from 1 l of culture). After 1 h of incubation, the suspension was transferred into a gravity flow column and the flow-through was discarded. The resin was washed with 30 column volumes of HBS (10 mM HEPES pH 7.5, 150 mM NaCl) supplemented with 30 mM imidazole and the protein was eluted with HBS supplemented with 300 mM imidazole. The elution was concentrated using centrifugal spin filters (Amicon, 10 kDa), and subjected to size-exclusion chromatography on a Superdex 75 10/300 column (GE Healthcare) in HBS buffer supplemented with 0.5 mM EDTA. Sybody-containing fractions were concentrated to >100 µM, flash-frozen in liquid nitrogen and stored at –80 °C until further use. For validation of the biochemical stability of sybodies obtained from HEK293T cells, expression and purification proceeded as described for the LRR domain of LRRC8A.

**Protein biotinylation**. For selection and ELISA screening, purified full-length LRRC8A, the LRRC8A-PD and LRR domains of LRRC8A, C and D were chemically biotinylated using amine-reactive EZ-Link™ NHS-PEG4-Biotin (Thermo-Fisher Scientific). For this purpose, proteins were diluted to 0.5 mg ml⁻¹ in their appropriate storage buffers and the coupling agent was added at sevenfold molar excess. The samples were incubated for 1 h on ice and the reaction was quenched by addition of 5 mM Tris-HCl, pH 8.0. Full-length LRRC8A and the LRRC8A-PD was subjected to size-exclusion chromatography using a Superose6 10/300 column (GE Healthcare) in HBS buffer supplemented with 100 µM GDN. Peak fractions were collected, supplemented with 20 µM soy polar extract lipids (Avanti), aliquoted and flash-frozen. LRR domains of LRRC8A, C and D were subjected to size-exclusion chromatography on a Superdex 75 10/300 column (GE Healthcare) in HBS buffer. Peak fractions were pooled, aliquoted and flash-frozen in liquid nitrogen. Samples were stored at –80 °C and aliquots were used immediately after thawing.

**Sybody selection and ELISA screening**. Sybody selection was performed using three synthetic ribosome-display libraries, each encoding for more than 10¹²

unique clones, referred to as concave (S), loop (M) and convex library (L). Sybodies were selected against chemically biotinylated full-length LRRC8A following a previously described protocol[30,31]. In brief, one round of ribosome-display was performed and the output sub-library was cloned into a phage-display compatible vector. These sub-libraries were subsequently used for two consecutive rounds of phage-display selection. In the second round an off-rate selection step was performed by addition of 5 µM of non-biotinylated LRRC8A for one minute to remove binders with fast dissociation rates. Phagemids of the output phages of the second round were isolated and the corresponding sybodies were subcloned using FX cloning into expression vector pSBinit, followed by single-clone screening by ELISA. The phage count in the elution fractions was monitored using qPCR and used for determination of enrichment factors. Briefly, during each round of phage display, parallel control pannings were performed using biotinylated AcrB protein as a target. Enrichment factors were calculated as ratios of phage titers in the elution fractions of LRRC8A and AcrB. The reported enrichment factors after the first and second round were 3.1 and 270.8, 1.6 and 52.3, 1.5 and 34.9 for concave, loop and convex libraries, respectively. All binding steps during selection and ELISA were performed in the appropriate buffers supplemented with 20 µM of soy polar lipids (Avanti) and 100 µM GDN. Wash buffers were supplemented with 100 µM GDN. To confirm binding specificity, a secondary ELISA was performed, which included more target proteins (full-length LRRC8A, LRRC8A-PD and the LRR domain of LRRC8A). Positive clones were sequenced. Based on ELISA results, 5 sybodies binding to the LRR domain of LRRC8A but not to LRRC8A-PD were selected for further analysis which are referred to as Sb1^LRRC8A (Sb1), Sb2^LRRC8A (Sb2), Sb3 ^LRRC8A (Sb3), Sb4 ^LRRC8A (Sb4) and Sb5 ^LRRC8A (Sb5).

**Binding assays by SEC and surface plasmon resonance**. For the analysis of complex formation by size-exclusion chromatography, the LRR domains of LRRC8A, C and D were mixed with purified sybodies in HBS supplemented with 0.25 mM DDM at a 1:3 molar ratio to obtain final LRR domain concentrations of 0.5 mg ml⁻¹ (~10 µM) and mixtures were incubated for 15 min on ice. 25 µl of the samples were injected onto a Superdex 200 5/150 column (GE Healthcare), eluted with HBS supplemented with 0.25 mM DDM at a flow rate of 0.2 ml min⁻¹. Peak fractions were collected and analyzed by SDS-PAGE. For the analysis by SPR, interactions between the LRR domain of LRRC8A and sybodies were analyzed using a Biacore T100 instrument (GE Healthcare). The chemically biotinylated LRR domain of LRRC8A was immobilized on a streptavidin-coated sensor chip (Cytiva) to obtain a maximum value of about 100 response units (RU). Measurements were performed at 20 °C at a flow rate of 30 µl min⁻¹ in 10 mM HEPES pH 7.4, 150 mM NaCl, 0.5 mM EDTA and 0.05% Tween 20. For sybodies Sb1, Sb5, Sb3 and Sb4, multi-cycle kinetics measurement with eight concentrations of the analyte were performed (1000, 500, 250, 125, 62.5 31.25, 15.63 and 7.81 nM). Due to its slow dissociation rate, the interaction with sybody Sb2 was characterized using single-cycle kinetics with injection of six concentrations of the analyte (1000, 500, 250, 125, 62.5, and 31.25 nM). Traces from a channel not containing immobilized protein and from an injection not containing an analyte were used as double reference for sensograms. Data processing and analysis were performed using BiaEvaluation software. Kinetic parameters were fitted using either 1:1 or heterogenous ligand interaction models.

**Surface expression analysis**. For the analysis of surface expression of endogenous LRRC8 channels, HEK293T cells were cultured in high glucose DMEM (Gibco), supplemented with 10% FBS, 4 mM glutamine, 1 mM sodium pyruvate and 100 U ml⁻¹ Penicillin-Streptomycin (Sigma) at 37 °C and 5% CO₂. Cells at 60% confluency were transfected with 10 µg of pcDXc3VMS plasmids encoding the respective sybodies per 10 cm dish using 25 µg of 40 kDa linear polyethyleneimine (Polysciences) and grown for 24 h. For biotinylation, cells from a 10 ml culture were washed with PBS and plasma membrane proteins were labeled with 10 ml Sulfo-NHS-SS Biotin solution with a concentration of 0.25 mg ml⁻¹ using the Pierce™ Cell Surface Protein Isolation Kit (ThermoFisher Scientific). For non-biotinylated control, cells were incubated with PBS instead. After 15 min at room temperature, biotinylation was stopped by addition of 500 µl of quenching solution and cells were harvested. Extraction and purification steps were carried out at 4 °C. After washing with cold PBS, the cell pellet was resuspended in 200 µl of the extraction buffer containing 10 mM HEPES pH 7.5, 150 mM NaCl, 1% GDN, 1x cOmplete EDTA-free Protease Inhibitors (Roche) and 10 µg ml⁻¹ DNAse and incubated for 1 h under gentle agitation. Insoluble fractions were removed by centrifugation at 14,000 g for 15 min. The supernatant was mixed with 100 µl of bed volume NeutrAvidin Agarose resin equilibrated in wash buffer (10 mM HEPES pH 7.5, 150 mM NaCl, 0.03% GDN) and incubated for 1 h under gentle agitation. The flow-through was discarded and the resin was washed three times with 200 µl of wash buffer. Captured proteins were eluted from the resin by incubation with wash buffer supplemented with 50 mM of fresh DTT for 1 h. For SDS PAGE, the total protein concentration was normalized based on the absorption at 280 nm (A₂₈₀). 5 µl at an A₂₈₀ of 0.1 were mixed with loading buffer and used for analysis. After separation by SDS-PAGE, proteins were transferred to a PVDF membrane and analyzed by Western blot using a mouse anti-LRRC8A primary antibody (Sigma, SAB1412855, 1:1000 dilution), a mouse anti-pan-cadherin primary antibody as loading control (abcam, ab6528, 1:1000 dilution) and a peroxidase-conjugated goat anti-mouse secondary antibody (Jackson ImmunoResearch, 115-

035-146, 1:10,000 dilution). The luminescent signal was developed with ECL substrate (GE Healthcare) and recorded with a Viber Fusion FX7 imaging system. Specific bands were quantified using Fiji[48]. For quantification, the signal corresponding to LRRC8A was normalized to the signal of pan-cadherin for each sample.

**Cryo-EM sample preparation and data collection.** Samples for structure determination of LRRC8A/sybody complexes were prepared by addition of respective sybodies from a highly concentrated stock to a purified and concentrated sample of LRRC8A without any following chromatography steps. Samples were concentrated to a final LRRC8A concentrations of 3–4 mg ml$^{-1}$ with a molar ratio of sybody/LRRC8A monomer of 1.25–1.5. For the sample LRRC8A-Sb4$_{0.5}$, the sybody was added sub-stoichiometrically (at a molar ratio of sybody/LRRC8A monomer of 0.5). For vitrification, 2.5 μl of protein samples were applied to glow-discharged holey carbon grids (Quantifoil R1.2/1.3 Au 200 mesh). Excess of the samples was removed by blotting grids for 3–7 s with 0 blotting force in a controlled environment (4 °C and 100% relative humidity). Grids were flash-frozen in a mixture of liquid ethane/propane using a Vitrobot Mark IV (ThermoFisher Scientific). Samples were imaged on a 300 kV Titan Krios G3i (ThermoFisher Scientific) with a 100 μm objective aperture. All data were collected using a post-column BioQuantum energy filter (Gatan) with a 20 eV slit and a K3 Summit direct detector (Gatan) operating in super-resolution mode. Dose-fractionated micrographs were recorded with a defocus range of −1.0 to −2.4 μm in an automated mode using EPU 2.5 (ThermoFisher Scientific). Data was recorded at a nominal magnification of ×130,000 corresponding to a pixel size of 0.68 Å/pixel (0.34 Å/pixel in super-resolution) with a total exposure time of 1 s (36 individual frames) and a dose of ~1.69 e$^{-}$/Å$^2$/frame. The total electron dose on the specimen level for all datasets was ~61 e$^{-}$/Å$^2$. The pixel size was later refined to the value of 0.651 Å/pixel (0.326 Å/pixel in super-resolution) and therefore the actual electron dose per frame and total dose were 1.85 e$^{-}$/Å$^2$/frame and 67 e$^{-}$/Å$^2$, respectively.

**Cryo-EM image processing.** In total, six datasets of LRRC8A in complex with five different sybodies were collected. All data processing was performed in Relion 3.0.8 and Relion 3.1 (ref. [49,50]) by a similar general procedure described below. Detailed information and processing steps relevant to a specific dataset are included in Supplementary Figs. 4–8. In all datasets, acquired super-resolution images were gain-corrected and down-sampled twice using Fourier cropping resulting in a pixel size of 0.68 Å. All frames were used for beam-induced movement correction with dose-weighting scheme using MotionCor2 (ref. [51]). CTF parameters were estimated using CTFFIND4.1 (ref. [52]). Micrographs showing a large drift, high defocus or poor CTF estimates were removed. Particles were auto-picked using templates imported from previously reported dataset of full-length LRRC8A[18]. Particles were extracted with a box size of 672 pixels (457 Å) and compressed four times (168-pixel box size, 2.7 Å/pixel) for initial processing. Extracted particles were subjected to one or two rounds of reference-free 2D classification followed by one or two rounds of 3D classification with C1 symmetry. During the first iteration of 3D classification, a previously determined map of LRRC8A[18] was used as reference after low-pass filtering to 60 Å. In further processing steps, the respective best maps at each stage were used as references after low-pass filtering to 40 Å. The resulting selected particles were re-extracted with twofold binning (336-pixel box size, 1.36 Å/pixel). Particles were then subjected to 3D auto-refinement with masks encompassing only protein density and excluding the density of the detergent micelle. To improve resolution, each pool of the final particles were for each dataset subjected to three rounds of iterative 3D auto-refinement, per-particle CTF correction[49] and single-particle motion correction[53]. Polished particles were used to generate final global reconstructions followed by masked local refinement of assemblies of the pore and cytosolic domains with bound sybodies, with either C3 or, in case of the pore domain of the LRRC8A/Sb2 dataset, C6 symmetry applied. For datasets of LRRC8A with sybodies Sb2, Sb4, and Sb4$_{0.5}$, local refinement of the regions encompassing cytosolic domains regions did not improve the resolution of this region (Supplementary Figs. 4 and 6). In contrast, for datasets LRRC8A/Sb1 and LRRC8A/Sb3, it was possible to improve the resolution of the cytosolic domains by subjecting the particles to symmetry expansion, signal subtraction and local refinement using masks including only a dimer of LRR domains and their bound sybodies and applying C1 symmetry. All acquired maps were sharpened using isotropic b-factors and the resolution was estimated using a soft solvent mask and based on the gold standard Fourier Shell Correlation (FSC) 0.143 criterion[54–57]. Final reconstructions were adjusted to the recalibrated pixel size of 0.651 Å/pixel by performing two rounds CTF refinement followed by map sharpening using the refined pixel size to scale the maps accordingly.

**Model building and refinement.** The models of full-length LRRC8A were based on a previously determined structure (PDB entry 6G9O)[18]. Initially, the hexameric model was fitted into the cryo-EM density using UCSF Chimera[58]. Subsequently, the pore (residues 15–411) and cytosolic domains (residues 412–808) of each protomer were fitted separately as rigid bodies into the density using Coot[59]. Initial models of respective sybodies were generated with the Swiss-model homology modeling server[60] based on templates of previously reported crystal structures of nanobodies (PDBIDs 3K1K[61] for Sb1 and Sb5, 3P0G[62] for Sb3 and 1ZVH[63] for

Sb4). Sb2 was modeled based on the refined structure of the related sybody Sb3. Sybodies were fitted into their corresponding densities using UCSF Chimera. Atomic models were further improved using rigid-body-fitting and real-space refinement in Coot and Phenix[64,65]. In all cases, except for the low-resolution dataset LRRC8A/Sb4, it was possible to place and refine pore-domain residues 15–68, 92–176, and 230–411 reliably. In case of datasets LRRC8A/Sb1 and LRRC8A/Sb3 it was also possible to unambiguously place residues 412–810 of the channel located in the LRR domain and residues 1–116 of the sybody Sb1 and 1–120 of the sybody Sb3. In data of complexes of the other three sybodies (Sb2, Sb4, and Sb5), models were refined using reference model restrains of previously reported LRRC8A structures and homology models of the sybodies[18]. In structures based on homology models, the protein-sybody interactions are tentative. Figures and movies containing molecular structures and densities were prepared with DINO (http://www.dino3d.org) and Chimera[58]. Surfaces were generated with MSMS[66].

**Electrophysiology.** Cells used for electrophysiological measurements were cultured in high glucose DMEM (Gibco), supplemented with 10% FBS, 4 mM glutamine, 1 mM sodium pyruvate and 100 U ml$^{-1}$ Penicillin-Streptomycin (Sigma) at 37 °C and 5% CO$_2$. To investigate the effect of the sybody expression on endogenous VRAC currents, HEK293T cells were gently detached from their support and seeded in 10-cm Petri dishes at 5% confluency. After a 2 h incubation step (to allow cells to adhere), cells were transfected with 3 μg of plasmid DNA encoding for sybodies fused to a Venus fluorescent tag using Lipofectamine 2000 transfection reagent (Invitrogen). For characterization of the effect of sybodies on the activity of heterologously expressed LRRC8A, HEK293 LRRC8-knockout cells (LRRC8$^{-/-}$, provided by T. J. Jentsch) were used. LRRC8$^{-/-}$ cells were cultured as described for HEK293T cells. One day before the measurement, cells were split by trypsinization and transfected after 2 h with 6 μg of DNA mixture (LRRC8A:sybody, 1:1) using Lipofectamine 2000. Whole-cell currents were measured 20–28 h after transfection. The effect of sybody Sb1 applied via the pipette solution on native VRAC currents was investigated on non-transfected HEK293T cells. All measurements were performed at 20 °C. Patch pipettes were pulled from borosilicate glass capillaries with inner diameter of 0.86 mm and outer diameter of 1.5 mm. The typical pipette resistance was 3–6 MΩ when filled with intracellular solution (125 mM salt). Seals with resistance of 4 GΩ or higher were used to establish the whole-cell configuration. Series resistance was compensated by 60% and was most commonly between 1 and 6 MΩ after compensation. The recordings were performed using Axopatch 200B and either Digidata 1440 or 1550 (Molecular devices). Analogue signals were digitized at 10–20 kHz and filtered at 5 kHz using the in-built 4-pole Bessel filter. Data acquisition was performed using Clampex 10.6 software (Molecular devices). Cells were locally perfused using gravity-fed system. Liquid junction potentials were corrected when it was calculated (in Clampex) to exceed 2 mV. Solutions used for recording were iso-osmotic with an osmolarity of 310–332 mmol kg$^{-1}$, as measured with a vapor pressure osmometer (VAPRO). The standard pipette solution for activation at decreased intracellular salt concentration was composed of 10 mM HEPES-NMDG pH 7.40, 1 mM EGTA, 2 mM Na$_2$ATP, 125 mM NMDG-Cl, 50 mM D-mannitol. The absence of divalent cations in the intracellular solution and presence of the Ca$^{2+}$ chelator EGTA and the Mg$^{2+}$ chelator Na$_2$-ATP potentiates channel activity even at mildly decreased salt concentrations[18]. When characterizing the effect of Sb1 included in the intracellular solution, the inside-buffer was supplemented with 1 μM of the sybody. For measurements at lower ionic strength, the NMDG-Cl concentration was decreased and the osmolarity was compensated by addition of mannitol. Unless stated otherwise, the external solution was composed of 10 mM HEPES-NMDG pH 7.4, 145 mM NaCl, 1.8 mM CaCl$_2$ and 0.7 mM MgCl$_2$. For measurements at high ionic strength, the salt concentration in both intracellular and extracellular solutions was increased to balance the osmolarity. After break-in into the cell and establishment of the whole-cell configuration, activation of VRAC currents was followed in 2 s intervals for 5–7 min using a ramp protocol (15 ms at 0 mV, 100 ms at −100 mV, a 500 ms linear ramp from −100 to 100 mV, 100 ms at 100 mV, 200 ms at −80 mV, 1,085 ms at 0 mV). The values at 100 mV, 10 ms after the ramp are displayed in the activation curves. Current-voltage relationships (I–V) were obtained from a voltage-jump step protocol (from −100 to 120 mV in 20 mV steps). Current rundown was corrected using a pre-pulse recorded at −80 mV preceding each voltage ramp. Swelling activated currents were measured as described[8]. The intracellular solution contained 10 mM HEPES-CsOH pH 7.2, 40 mM CsCl, 100 mM Cs-methanesulfonate, 1 mM MgCl$_2$, 1.9 mM CaCl$_2$, 5 mM EGTA freshly supplemented with 4 mM Na$_2$ATP-NMDG pH 7.2. Cells were perfused with isotonic buffer (10 mM HEPES-NaOH pH 7.4, 150 mM NaCl, 6 mM KCl, 1 mM MgCl$_2$, 1.5 mM CaCl$_2$, 10 mM glucose, 320 mOsm) for 120–150 s after establishing the whole-cell configuration and swelling was initiated by switching the perfusion buffer to hypotonic buffer (10 mM HEPES-NaOH pH 7.4, 105 mM NaCl, 6 mM CsCl, 1 mM MgCl$_2$, 1.5 mM CaCl$_2$, 10 mM glucose, 240 mOsm). Currents were monitored in 2 s intervals for 4–5 min using a ramp protocol described above. Current-voltage relationships were obtained and analyzed as described above. For the measurements in hypotonic conditions only one cell was used per dish. Data was analyzed using Clampfit 10.6 (Molecular devices), Excel (Microsoft) and GraphPad Prism 8.

**Statistics and reproducibility**. Electrophysiology data were repeated multiple times from different transfections with very similar results. Conclusions of experiments were not changed upon inclusion of further data. In all cases, leaky patches were discarded.

**Reporting summary**. Further information on research design is available in the Nature Research Reporting Summary linked to this article.

## Data availability

The three-dimensional cryo-EM density maps have been deposited in the Electron Microscopy Data Bank under accession numbers EMD-13202 (LRRC8A/Sb1), EMD-13203 (LRRC8A/Sb2), EMD-13208 (LRRC8A/Sb3), EMD-13212 (LRRC8A/Sb4), EMD-13213 (LRRC8A/Sb4$_{0.5}$) and EMD-13230 (LRRC8A/Sb5). The deposition includes maps of full-length proteins, corresponding half-maps 1 and 2, the mask used for final FSC calculation as well as relevant higher resolution maps obtained after local refinement. Coordinates for the models of full-length LRRC8A/Sb1, LRRC8A/Sb2, LRRC8A/Sb3, LRRC8A/Sb4$_{0.5}$ and LRRC8A/Sb5 have been deposited in the Protein Data Bank under accession numbers 7P5V [https://doi.org/10.2210/pdb7P5V/pdb], 7P5W [https://doi.org/10.2210/pdb7P5W/pdb], 7P5Y [https://doi.org/10.2210/pdb7P5Y/pdb], 7P60 [https://doi.org/10.2210/pdb7P60/pdb] and 7P6K [https://doi.org/10.2210/pdb7P6K/pdb], respectively. The data from electrophysiological recordings showing the effect of sybodies on LRRC8 currents have been deposited in the Dryad database (https://doi.org/10.5061/dryad.ht76hdrgg). Source data are provided with this paper.

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

## Acknowledgements

This research was supported by grants from the Swiss National Science Foundation (No. 310030B_182828 to R.D. and No. 310030_188817 to M.A.S.). We thank Simona Sorrentino and the Center for Microscopy and Image Analysis (ZMB) of the University of Zurich for the support and access to the electron microscopes, Iwan Zimmermann for help in sybody selection, Jens Sobek from the Functional Genomics Center of the UZH/ETH Zurich for help with surface plasmon resonance experiments and T. J. Jentsch for providing the *LRRC8⁻/⁻* HEK cell-line and an aliquot of an Anti-LRRC8A antibody used for preliminary studies. All members of the Dutzler lab are acknowledged for their help at various stages of the project. The cryo-electron microscope and K3-camera were acquired with support of the Baugarten and Schwyzer-Winiker foundations and a Requip grant of the Swiss National Science Foundation (No. 316030_183382).

## Author contributions

D.D. and S.R. generated expression constructs and purified proteins. D.D. and C.A.J.H. selected sybodies and M.A.S. supervised sybody selection. S.R. carried out surface-biotinylation experiments. M.S. oversaw cryo-EM experiments, M.S. and D.D. and S.R. prepared the samples for cryo-EM and collected cryo-EM data. D.D., S.R., and M.S. proceeded with structure determination and refinement. D.D. recorded and analyzed electrophysiology data. D.D., S.R., M.S., and R.D. jointly planned experiments, analyzed the data and wrote the manuscript. M.A.S. edited the manuscript.

## Competing interests

The authors declare no competing interests.
