## [Peer Review File · Nature Communications]

Allosteric modulation of LRRC8 channels by targeting their cytoplasmic domainsREVIEWER COMMENTS

Reviewer #1 (Remarks to the Author):

This is a paper that will be of interest to experts in the VRAC field. However, at this early stage the paper's impact is reduced by the lack of novel insights into the molecular basis of VRAC activation.

MAJOR:

1. The authors focus on using changes in intracellular ionic strength to modulate channel activity. Changes in ionic strength are not the primary signal for VRAC activation/inactivation. Intracellular ionic strength modulates the volume sensitivity of VRAC. How cell volume changes mediate changes in channel activation remain unknown. This begs the question of whether sybodies have similar effects on swelling-induced channel activation and shrinkage-induced inactivation. Also of concern, albeit not addressable at this time, is whether activation by low ionic strength occurs via the same molecular mechanisms as volume dependent activation/inactivation.

2. I'm very confused by the electrophysiology data presented. In Figure 2 and extended Figure2 the authors show that currents are activating in cells equilibrated with a 125 mM salt solution. This is equivalent to a "normal" intracellular ionic strength. No activation should be occurring until intracellular salt levels are reduced to very low and unphysiological levels, typically close to or below 75 mM. Without measurements of cell volume, are the authors certain that their bath and pipette solutions are balanced correctly to avoid spontaneous (i.e., in the absence of changes in bath osmolarity) volume perturbations? This is a serious concern. The absence of concomitant cell volume and electrophysiology measurements contributed in part to the extensive early confusion in the VRAC field. Are the sybodies in a salt solution and does their addition change pipette osmolarity?

3. In the methods section the authors state, "for patch-clamp experiments full-length LRRC8A and LRRC8C were cloned into an analogous vector not containing mCherry (pcDXc3MS)". Unless I missed it, no electrophysiology data are presented for heterologously expressed LRRC8A/C heteromeric channels. Were any studies with sybodies conducted on heterologously expressed heteromeric channels?

4. A very significant shortcoming is the inability to correlate observed conformational changes to channel activity. No mutagenesis-based structure/function analyses are performed. Unfortunately, the methods used in this paper complicate such studies. As the authors note, LRRC8A anion channels likely do not exist in Nature and have non-native functional properties. Studies by Kern et al. illustrate the problem of translating cryo-EM structures into a molecular understanding of VRAC structure/function relationships. Kern et al. recently solved the cryo-EM structure of LRRC8A blocked by DCPIB. However, as shown by Yamada et al. DCPIB inhibition of LRRC8A bears no resemblance to that of native VRACs. Furthermore, an amino acid residue implicated by Kern et al. in LRRC8A DCPIB inhibition is not important for inhibition of chimeric or heteromeric LRRC8 channels. Studies of heteromeric channels are complicated by the lack high confidence, high resolution structures and by the unknown and uncontrollable stoichiometry and assembly order of LRRC8 heteromers.

5. The conclusion that the LRR domains function as a possible allosteric regulator of LRRC8 channels is not novel and is supported by multiple other published studies. As the authors know, it has been demonstrated in multiple proteins that LRR domain are highly flexible and play central roles in regulation of protein function. Evidence for regulatory conformational changes in the LRRC8 LRR domains was recently published by Konig et al. Kern et al. noted that extracellular DCPIB block alters the conformation of intracellular domains suggesting that there is allosteric communication between extracellular and intracellular regions of LRRC8 channels. As with some of the sybodies presented in this paper, addition of fluorescent tags to the C-termini of LRRC8 proteins constitutively activates channels heterologously expressed in *Xenopus* oocytes (Gaitan-Penas et al.)

MINOR:

1. The authors note that the sybodies do not interact with the LRR domains of LRRC8C and LRRC8D. This is surprising given the conservation of primary sequence and suggests a novel structure of the LRRC8A LRR domain. This issue should be discussed briefly in the manuscript.
2. The authors refer to CLH1 and CLH2. Not all readers may be familiar with this terminology. A 2D figure of the LRRC8 subunit would be helpful.

Reviewer #2 (Remarks to the Author):

This manuscript describes the development of synthetic nanobodies (sybodies) that target the LRR domain of the LRRC8A channel. The authors show that the sybodies modulated (activated or inhibited) the activity of the channel, thereby establishing the regulatory roles of the LRR domain. They determined the cryo-EM structures of the LRRC8A homo-hexamers in complex with the sybodies, which revealed the epitopes and potential mechanisms of channel regulation by the sybodies. Overall, this is a very nice study that should appeal to broad audience.

The experiments are well designed, executed and described. The sybodies represent novel and powerful tools that can enable future mechanistic studies of these interesting channels.

The main weakness of this work is the challenge in connecting the structural studies performed using the LRRC8A homo-hexamers to the mechanism of the physiological LRRC8A channel that forms heteromeric complexes of unknown stoichiometry of LRRC8 subunits. The proposed mechanism in which the sybodies alter the stability and/or dynamics of the LRR hexamer is reasonable and supported by the functional data of the LRRC8A homohexamer. However, it is also possible that these sybodies function, under physiological conditions, by disrupting interactions between the LRR with yet-to-be-identified ligands. Also, it is not clear whether this mechanism applies to the physiologically relevant heteromeric complexes. These are important points that the authors should expand on, perhaps by discussing sequence conservation among different subunits and information available in the literature. Related to this issue is that Discussion essentially repeats Results and it does not place this work in a larger context. Most of the current text under Discussion can be omitted. Instead, I would like to see more nuanced discussion of this work including the limitations listed above.

It would be nice to see mutation studies of LRR residues in the sybody epitopes, which may further support the proposed mechanisms. However, I would not consider such studies essential for publication.

Figures 3-6, as separately presented, are not as effective as they can be. A major strength of this work is the series of the structures with different sybodies, which allows for nice comparisons. I found Extended Data Figure 9 to be more informative than Figures 3-6. I encourage the authors to rearrange these figures so that they can more effectively illustrate the most important points.

Extended Data Fig. 2a and 2b. "S33" a typo?

Reviewer #3 (Remarks to the Author):

Allosteric modulation of LRRC8 channels by targeting their cytoplasmic domains

In 2018, Deneka et al. reported the architecture of the homomeric LRRC8A channel for the first time. In this current work, the authors set out to understand the role of the cytoplasmic LRR domain in the activation and regulation of the LRRC8 channels and utilized a combined technique of synthetic nanobodies (sybodies) and structural analysis by single-particle cryo-electron microscopy (cryo-EM). The authors generated five sybodies that specifically target the LRR domain of LRRC8A and characterized the respective activating or inhibitory roles of the sybodies on the LRRC8A channels. Furthermore, to understand the mechanism of the sybodies on channel function, the authors determined the cryo-EM structures of LRRC8A channel in complex with individual sybodies. They captured the distinct binding modes of the sybodies in the LRR domains and revealed specific conformational changes in the LRRC8A channels induced by sybody binding, thereby suggesting the structural basis for the role of the LRR domain on the allosteric modulation of LRRC8 channels. This reviewer believes that the work by Deneka et al. exemplifies high-quality structural studies and mechanistic understanding of ion channel functions combined with the usage of synthetic nanobody technology. A few minor points considering the cryo-EM data presentation and analysis are listed below.

1. In Extended Data Fig. 3-7, the authors should provide the FSC curves for structural models against EM full-map and both half-maps.

2. For structure determination of apo LRRC8A, the pore domain was masked auto-refined with C6 symmetry, and the final full-length protein was assembled from the pore domain and crystal structure of the LRR domain into a C3-symmetry EM map (Deneka et al., 2018). In contrast, in this manuscript, the 3D-reconstructions of LRRC8A channel in complex with different sybodies display reduced symmetry (C3) in channel transmembrane domain and/or the cytoplasmic LRR domains. For cryo-EM data processing, the authors started without symmetry imposed (C1) in initial 3D classifications, impose C3 symmetry in further 3D refinement of the full complex, or impose C3 or C6 symmetry in focused refinement of specific domains. In Extended Data Fig. 3-7, the authors showed the side and bottom views of low-resolution 3D classes, but the symmetry is not clear. The authors should provide justification of the basis of applying symmetry. For instance, they may show cross-section views for alignment of the 3D reconstructions of LRRC8A/sybody complex and apo LRRC8A and show deviation from C6 symmetry in the pore domain and the LRR domain.

3. In main text and in Methods (page 41), the authors built models for Sb2/4/5 using the homology models due to the low local-resolution of the sybody EM density in the corresponding 3D reconstructions. To strengthen the structural interpretations of LRR/sybody interaction and the sybody action on the channel function in the main text, the authors may need to show superimposition of EM density and key structural features for sybodies 2/4/5 and Sb/LRR domain interfaces.

4. The authors performed focus refinement and/or signal subtraction for the pore domain, LRR domain, the LRR dimer for the complex 3D reconstructions. The reviewer assumes that for EMDB deposition, the authors deposited the EM maps for the full channel/sybody complex, not the local refined higher-resolution maps. For each channel/sybody complex structure, which map(s) did the authors actually use for model building? Did they always use separate high-resolution local EM maps (e.g., the pore domain or the LRR domains) for model building then place a "chimeric" structure coordinates into the full channel/sybody map?

5. In Methods, page 39, given the ~600kDa molecular weight of the full-length LRRC8A channel, is there a particular reason that the cryo-EM data was collected with a small pixel size of 0.68Å/pixel?

We thank the reviewers for their generally positive and constructive comments. Following their suggestions, we have introduced several changes to the manuscript and provide a detailed response to their comments below.

Reviewer #1 (Remarks to the Author):

This is a paper that will be of interest to experts in the VRAC field. However, at this early stage the paper's impact is reduced by the lack of novel insights into the molecular basis of VRAC activation.

MAJOR:

1. The authors focus on using changes in intracellular ionic strength to modulate channel activity. Changes in ionic strength are not the primary signal for VRAC activation/inactivation. Intracellular ionic strength modulates the volume sensitivity of VRAC. How cell volume changes mediate changes in channel activation remain unknown. This begs the question of whether sybodies have similar effects on swelling-induced channel activation and shrinkage-induced inactivation. Also, of concern, albeit not addressable at this time, is whether activation by low ionic strength occurs via the same molecular mechanisms as volume dependent activation/inactivation.

To address the concern related to the activation protocol of VRACs, we performed whole-cell patch-clamp electrophysiology measurements of swelling-activated VRAC currents (as described by Voss et al.¹) in non-transfected HEK293T cells and the same cells transfected with either a sybody that does not target LRRC8 proteins (Sbn) or the inhibitory sybody Sb1. Our data shows a strong reduction of the observed currents upon transfection with Sb1, but not with the control sybody Sbn thus demonstrating that the inhibitory effect is independent of the activation protocol. The data has been included in our revised manuscript as Supplementary Fig. 2.

Although we agree that the signals leading to VRAC activation in a physiological context are currently not understood, we and others have previously found that the channels can be reliably activated by a decrease of the intracellular ionic strength in combination with a reduction of the divalent cation concentration^{2,3}. We have thus used this approach in this study to investigate the influence of sybodies on channel activation. Since both activation protocols result in comparable general properties, it appears likely that the respective activated states share very similar features.

2. I'm very confused by the electrophysiology data presented. In Figure 2 and extended Figure2 the authors show that currents are activating in cells equilibrated with a 125 mM salt solution. This is equivalent to a "normal" intracellular ionic strength. No activation should be occurring until intracellular salt levels are reduced to very low and unphysiological levels, typically close to or below 75 mM. Without measurements of cell volume, are the authors certain that their bath and pipette solutions are balanced correctly to avoid spontaneous (i.e., in the absence of changes in bath osmolarity) volume perturbations? This is a serious concern. The absence of concomitant cell volume and electrophysiology measurements contributed in part to the extensive early confusion in the

VRAC field. Are the sybodies in a salt solution and does their addition change pipette osmolarity.

Solutions and patch-clamp protocols were already used in a previous study². As described in the methods section of this paper, the osmolarities of all buffers were confirmed experimentally using a vapor pressure osmometer (line 646-648). The sybodies were purified in 10 mM HEPES pH 7.5, 150 mM NaCl, 0.5 mM EDTA and concentrated to 200-300 μ M. The volume of the concentrated sybody added to the intracellular solution amounted to maximally 0.5% of the total volume and thus did not affect the osmolarity of the intracellular buffer. The contribution of the sybody itself to the osmolarity is likely negligible (1 μ M Sb1 is equivalent to 0.014 mg/ml).

Instead, we assume that the observed activating properties of intracellular buffer is influenced by the lack of divalent cations, such as Ca^{2+} and Mg^{2+} , and the high concentration of free ATP (2 mM), which was shown to potentiate VRACs⁴. We have previously found that in absence of ATP, addition of 1 mM MgCl_2 to the pipette solution strongly attenuates endogenous VRAC currents in HEK293T cells (Fig. 1), suggesting that the absence of Mg^{2+} (due to chelation by ATP) would in turn potentiate currents even at 125 mM salt.

Fig. 1 Inhibition of VRAC currents by intracellular Mg^{2+} . Currents were recorded by patch-clamp electrophysiology in the whole-cell configuration with a pipette solution containing 10 mM HEPES-NMDG pH 7.4, 100 mM NMDG-Cl, 1 mM EGTA and the indicated concentrations of Na_2ATP or Mg^{2+} . Bath solutions contained 10 mM HEPES-NMDG pH 7.4, 100 mM NMDG-Cl, 0.5 mM MgCl_2 , 0.5 mM CaCl_2 . Data show mean of five experiments, errors are s.e.m.

Although experiments were performed using osmotically balanced buffers, we cannot rule out a possible contribution of cell swelling or swelling-like phenomena to the activated currents.

3. In the methods section the authors state, “for patch-clamp experiments full-length LRRC8A and LRRC8C were cloned into an analogous vector not containing mCherry (pcDXc3MS)”. Unless I missed it, no electrophysiology data are presented for

heterologously expressed LRRC8A/C heteromeric channels. Were any studies with sybodies conducted on heterologously expressed heteromeric channels?

The text was added accidentally and was now removed from the methods.

4. A very significant shortcoming is the inability to correlate observed conformational changes to channel activity. No mutagenesis-based structure/function analyses are performed. Unfortunately, the methods used in this paper complicate such studies. As the authors note, LRRC8A anion channels likely do not exist in Nature and have non-native functional properties. Studies by Kern et al. illustrate the problem of translating cryo-EM structures into a molecular understanding of VRAC structure/function relationships. Kern et al. recently solved the cryo-EM structure of LRRC8A blocked by DCPIB. However, as shown by Yamada et al. DCPIB inhibition of LRRC8A bears no resemblance to that of native VRACs. Furthermore, an amino acid residue implicated by Kern et al. in LRRC8A DCPIB inhibition is not important for inhibition of chimeric or heteromeric LRRC8 channels. Studies of heteromeric channels are complicated by the lack high confidence, high resolution structures and by the unknown and uncontrollable stoichiometry and assembly order of LRRC8 heteromers.

Our study provides wealth of experimental data that describe the identification and characterization of the first set of specific modulators of VRAC function. Mutagenesis experiments are beyond the scope of the current work and will be subject of future studies.

Although we acknowledge in our manuscript that LRRC8A is a channel with compromised activation properties, its structural features are generally similar to VRAC heteromers, as shown in our previous publication². Moreover, our present study reveals equivalent functional effects of sybody interactions on homo- and heteromeric channels. This indicates that similar conformational changes leading to channel activation and inhibition are undergone by homo- and heteromeric channels. Finally, although of potential functional relevance, we clearly state that at this stage, we cannot assign observed conformations to a defined functional state of the protein (line 349-351).

With respect to the reference to DCPIB inhibition, we would like to emphasize the fundamental differences of the referred studies to our present work. Whereas DCPIB is a low-affinity, small molecule blocker of poor selectivity, which might potentially interact with multiple regions of the protein, the here described sybodies are subunit-specific and highly potent modulators that bind to single extended epitopes of the A subunit, as demonstrated by different biochemical and structural experiments described in our study. Although we expect that the structural features observed in homomeric channels might be enhanced (as stated in line 332-334), there is no reasonable doubt concerning the interaction with the LRR domain of LRRC8A, which is conserved in homo- and heteromeric channels.

5. The conclusion that the LRR domains function as a possible allosteric regulator of LRRC8 channels is not novel and is supported by multiple other published studies. As the authors know, it has been demonstrated in multiple proteins that LRR domain are highly flexible and play central roles in regulation of protein function. Evidence for regulatory conformational changes in the LRRC8 LRR domains was recently published by Konig et al.

Kern et al. noted that extracellular DCPIB block alters the conformation of intracellular domains suggesting that there is allosteric communication between extracellular and intracellular regions of LRRC8 channels. As with some of the sybodies presented in this paper, addition of fluorescent tags to the C-termini of LRRC8 proteins constitutively activates channels heterologously expressed in Xenopus oocytes (Gaitan-Penas et al.)

While we agree that the coupling between the LRR domains and the pore domain was proposed before, we believe that this manuscript presents the first extended study that provides direct evidence for the allosteric coupling and also offers structural insight into the process. We have now explicitly referred to the mentioned studies in the revised discussion (line 363-368).

MINOR:

1. The authors note that the sybodies do not interact with the LRR domains of LRRC8C and LRRC8D. This is surprising given the conservation of primary sequence and suggests a novel structure of the LRRC8A LRR domain. This issue should be discussed briefly in the manuscript.

While the sequence conservation between the LRRC8 proteins is fairly high (up to 65% identity for the murine LRRC8A and LRRC8C), even small differences in the amino-acid sequence can strongly affect the binding affinity without requiring large structural differences in the general architecture of the LRR domains as illustrated in Fig. 2. This is particularly the case for molecules of the immune system which are usually highly specific. Factors which could contribute to the LRRC8A selectivity include the local curvature and twist between the affected LRR domains, which will ultimately be shown in structures of other paralogs. It is thus not surprising to find the observed high subunit selectivity.

Fig. 2 Conservation of the Sb1 epitope residues between LRRC8A, C and D. Side chains within 5 Å of the sybody are shown as sticks. Red – residues conserved in all 3 paralogs, white – conserved in 2/3 of the paralogs, blue – present only in LRRC8A.

2. The authors refer to CLH1 and CLH2. Not all readers may be familiar with this terminology. A 2D figure of the LRRC8 subunit would be helpful.

We have added a figure of the LRRC8 subunit as part of panel g of Supplementary Fig. 11.

Reviewer #2 (Remarks to the Author):

This manuscript describes the development of synthetic nanobodies (sybodies) that target the LRR domain of the LRRC8A channel. The authors show that the sybodies modulated (activated or inhibited) the activity of the channel, thereby establishing the regulatory roles of the LRR domain. They determined the cryo-EM structures of the LRRC8A homo-hexamers in complex with the sybodies, which revealed the epitopes and potential mechanisms of channel regulation by the sybodies. Overall, this is a very nice study that should appeal to broad audience.

The experiments are well designed, executed and described. The sybodies represent novel and powerful tools that can enable future mechanistic studies of these interesting channels.

The main weakness of this work is the challenge in connecting the structural studies performed using the LRRC8A homo-hexamers to the mechanism of the physiological LRRC8A channel that forms heteromeric complexes of unknown stoichiometry of LRRC8 subunits. The proposed mechanism in which the sybodies alter the stability and/or dynamics of the LRR hexamer is reasonable and supported by the functional data of the LRRC8A homohexamer. However, it is also possible that these sybodies function, under physiological conditions, by disrupting interactions between the LRR with yet-to-be-identified ligands. Also, it is not clear whether this mechanism applies to the physiologically relevant heteromeric complexes. These are important points that the authors should expand on, perhaps by discussing sequence conservation among different subunits and information available in the literature. Related to this issue is that Discussion essentially repeats Results and it does not place this work in a larger context. Most of the current text under Discussion can be omitted. Instead, I would like to see more nuanced discussion of this work including the limitations listed above.

We have revised the discussion, extended the reference to previous work and discussed the limitations of the conclusions derived from the current work. The possibility that the sybodies might interfere with the interaction of a natural ligand is now explicitly mentioned (line 394-396). We have also added a novel figure for the discussion (Fig. 8)

It would be nice to see mutation studies of LRR residues in the sybody epitopes, which may further support the proposed mechanisms. However, I would not consider such studies essential for publication.

The characterization of sybody interaction by mutagenesis is beyond the scope of this manuscript and will be part of a future study.

Figures 3-6, as separately presented, are not as effective as they can be. A major strength of this work is the series of the structures with different sybodies, which allows for nice comparisons. I found Extended Data Figure 9 to be more informative than Figures 3-6. I

encourage the authors to rearrange these figures so that they can more effectively illustrate the most important points.

We have now added an additional figure to the manuscript (Fig. 3) where we show the cryo-EM densities of all five sybody complexes. The relationship between different epitopes is also illustrated in the new Fig. 8a.

Extended Data Fig. 2a and 2b. “S33” a typo?

This was corrected to ‘Sb1’.

Reviewer #3 (Remarks to the Author):

Allosteric modulation of LRRC8 channels by targeting their cytoplasmic domains

In 2018, Deneka et al. reported the architecture of the homomeric LRRC8A channel for the first time. In this current work, the authors set out to understand the role of the cytoplasmic LRR domain in the activation and regulation of the LRRC8 channels and utilized a combined technique of synthetic nanobodies (sybodies) and structural analysis by single-particle cryo-electron microscopy (cryo-EM). The authors generated five sybodies that specifically target the LRR domain of LRRC8A and characterized the respective activating or inhibitory roles of the sybodies on the LRRC8A channels. Furthermore, to understand the mechanism of the sybodies on channel function, the authors determined the cryo-EM structures of LRRC8A channel in complex with individual sybodies. They captured the distinct binding modes of the sybodies in the LRR domains and revealed specific conformational changes in the LRRC8A channels induced by sybody binding, thereby suggesting the structural basis for the role of the LRR domain on the allosteric modulation of LRRC8 channels. This reviewer believes that the work by Deneka et al. exemplifies high-quality structural studies and mechanistic understanding of ion channel functions combined with the usage of synthetic nanobody technology. A few minor points considering the cryo-EM data presentation and analysis are listed below.

1. In Extended Data Fig. 3-7, the authors should provide the FSC curves for structural models against EM full-map and both half-maps.

We now show the requested curves in Supplementary Fig. 9.

2. For structure determination of apo LRRC8A, the pore domain was masked auto-refined with C6 symmetry, and the final full-length protein was assembled from the pore domain and crystal structure of the LRR domain into a C3-symmetry EM map (Deneka et al., 2018). In contrast, in this manuscript, the 3D-reconstructions of LRRC8A channel in complex with different sybodies display reduced symmetry (C3) in channel transmembrane domain and/or the cytoplasmic LRR domains. For cryo-EM data processing, the authors started without symmetry imposed (C1) in initial 3D classifications, impose C3 symmetry in further 3D refinement of the full complex, or impose C3 or C6 symmetry in focused refinement of

specific domains. In Extended Data Fig. 3-7, the authors showed the side and bottom views of low-resolution 3D classes, but the symmetry is not clear. The authors should provide justification of the basis of applying symmetry. For instance, they may show cross-section views for alignment of the 3D reconstructions of LRRC8A/sybody complex and apo LRRC8A and show deviation from C6 symmetry in the pore domain and the LRR domain.

The referred side and bottom views in the cryo-EM workflow (panels d of Supplementary Figs 3-7) show the channel with no symmetry applied. In case of applied symmetry, this is mentioned explicitly. C3 and C6 symmetry was only applied if it resulted in an improvement of the map as illustrated in the increased resolution and improved features of the maps. For the TM domain of the LRRC8A-Sb2 complex, this was clearly the case, as its density improved considerably upon application of C6 symmetry. Although most of the TM domain of the LRRC8A-Sb1 complex exhibited C6 symmetry, there is a slight breakdown of symmetry in the intracellular sub-domain, close to the region connecting to the LRR domains. We thus refrained from applying C6 symmetry in this case and instead worked with a C3-symmetrized map. Due to the described changes in the TM domains of Sb2, Sb4 and Sb5 complexes, the symmetry reduction from C6 to C3 was evident in these cases. In case of the LRR domain, we restricted our detailed reconstruction to the fraction of particles showing C3-symmetry in this region. For the focused refinement of the domain pair there was obviously no symmetry applied since this part corresponds to the asymmetric unit of the C3-symmetric domain arrangement. The variable quality of the LRR reconstructions in different complexes is a consequence of their distinct degrees of flexibility and not a symmetry mismatch. Fig. 3 below shows examples of slices through the map. However, we do not find these slices sufficiently instructive (beyond what is shown elsewhere) to justify their inclusion as additional figure to the manuscript (which already contains wealth of supplementary data).

Fig. 3 Slices through the transmembrane domain of low-pass filtered, non-symmetrized maps of LRRC8A in complex with Sb1, Sb2 and Sb3. Top transmembrane region (TM), bottom, intracellular

subdomain (ISD). Regions showing pronounced differences between Sb1 and Sb3 complexes are indicated by asterisk.

3. In main text and in Methods (page 41), the authors built models for Sb2/4/5 using the homology models due to the low local-resolution of the sybody EM density in the corresponding 3D reconstructions. To strengthen the structural interpretations of LRR/sybody interaction and the sybody action on the channel function in the main text, the authors may need to show superimposition of EM density and key structural features for sybodies 2/4/5 and Sb/LRR domain interfaces.

We have now provided views of low-pass filtered maps (at 6Å) of the respective complexes to show the location of the sybody on the domain (Supplementary Fig. 1f, h, j). The assignment of the binding interface of Sb2 benefits from the strong overlap with Sb3, which is considerably better defined. Since the detailed structural features of interactions between Sb4 and Sb5 and the LRR domain cannot be derived from the data, we have removed panels f-j from the figure describing LRRC8A in complex with potentiating sybodies (previously Fig. 6 now Fig. 7).

4. The authors performed focus refinement and/or signal subtraction for the pore domain, LRR domain, the LRR dimer for the complex 3D reconstructions. The reviewer assumes that for EMDB deposition, the authors deposited the EM maps for the full channel/sybody complex, not the local refined higher-resolution maps. For each channel/sybody complex structure, which map(s) did the authors actually use for model building? Did they always use separate high-resolution local EM maps (e.g., the pore domain or the LRR domains) for model building then place a “chimeric” structure coordinates into the full channel/sybody map?

Parts of the models were initially built and refined into respective maps of highest available resolution and later combined and further refined using the map of the full-length complex. For the EMDB deposition, the cryo-EM map of the full channel/sybody complex will be deposited as a main map with other maps (after focus refinement and/or signal subtraction) deposited as auxiliary maps.

5. In Methods, page 39, given the ~600kDa molecular weight of the full-length LRRC8A channel, is there a particular reason that the cryo-EM data was collected with a small pixel size of 0.68Å/pixel?

A small pixel size is a consequence of the strategy that we adopted to collect the data. In order to boost the speed of data collection we decided to apply multi-shot targeting per hole using beam-image shift. Using high magnification and therefore a small pixel size, we were able to collect three exposures per hole ensuring there is no overlapping regions between the exposures.

1 Voss, F. K. *et al.* Identification of LRRC8 heteromers as an essential component of the volume-regulated anion channel VRAC. *Science* **344**, 634-638, doi:10.1126/science.1252826 (2014).

- 2 Deneka, D., Sawicka, M., Lam, A. K. M., Paulino, C. & Dutzler, R. Structure of a volume-regulated anion channel of the LRRC8 family. *Nature* **558**, 254-259, doi:10.1038/s41586-018-0134-y (2018).
- 3 Syeda, R. *et al.* LRRC8 Proteins Form Volume-Regulated Anion Channels that Sense Ionic Strength. *Cell* **164**, 499-511, doi:10.1016/j.cell.2015.12.031 (2016).
- 4 Bryan-Sisneros, A., Sabanov, V., Thoroed, S. M. & Doroshenko, P. Dual role of ATP in supporting volume-regulated chloride channels in mouse fibroblasts. *Biochim Biophys Acta* **1468**, 63-72, doi:10.1016/s0005-2736(00)00243-1 (2000).

REVIEWER COMMENTS

Reviewer #1 (Remarks to the Author):

The authors have addressed some of my concerns, but significant issues remain. I remain very concerned about the activation supposedly induced by changes in ionic strength. The authors now seem to be arguing that it is reduced divalent ion concentrations that activate the channel at normal ionic strength. So what is, reduced ionic strength, reduced divalents, a synergistic interaction between ionic strength and divalents or something else?

The suggestion that divalents play a regulatory role contradicts published studies. Multiple studies from several laboratories that are the leaders in the VRAC field have shown that reducing ionic strength has two effects. First, as ionic strength is reduced, less swelling is required to activate the channel. Second, when ionic strength is reduced to grossly unphysiological levels, VRAC activates without swelling. All of this occurs in the presence of constant levels of divalent ions.

The presence of ATP in the cell is an absolute requirement for activation unless the rate of cell swelling is extremely high. The requirement for ATP occurs in the presence or absence of divalent ions and ATP can be replaced by nonhydrolyzable analogs. The authors present unpublished data in their rebuttal that contradict these peer-reviewed published studies.

My criticism about drawing excessive conclusions from the LRRC8A structure remains. There can be no question that the LRRC8A homomer structure represents an important breakthrough in the field. However, we now know that LRRC8A homomers have grossly abnormal regulation and functional properties. Thus, drawing structural conclusions from LRRC8A homomers about how VRAC is regulated without molecular confirmation is unwarranted.

I don't believe the authors can state that the structural features of LRRC8A "are generally similar to VRAC heteromers" based on their work that has not yet been reproduced by other labs. The heteromeric structure generated by the Dutzler lab is low resolution, shows only a side view of the channel, and was reconstructed using the homomeric LRCC8A structure as a reference map and imposed C3 symmetry, both of which likely imposed a bias in the solved heteromeric structure. Furthermore, this low-resolution structure does not take into account the possibility that subunit stoichiometry could very well influence overall channel conformation.

Reviewer #2 (Remarks to the Author):

The authors have adequately addressed the points that I raised for the previous version of this manuscript.

Reviewer #3 (Remarks to the Author):

The revision is substantially improved, and the authors have satisfactorily addressed my minor technical points in their revision. Although there are apparent shortcomings of this study: the unclear functional states of sybody-bound LRRC8A structures and the use of physiologically less relevant homomeric LRRC8A for structural studies, I believe that this is the first study to show subunit-specific allosteric modulators-dependent conformational changes of the LRRC8 channel. Based on the fact that current level of mechanistic understanding of this channel family is very limited due to many technical hurdles, I believe that the contribution of this study to the field is significant enough to warrant publication to Nature Communications.

Our response to the comments of reviewer one and corresponding changes in the manuscript are provided below.

Reviewer #1 (Remarks to the Author):

The authors have addressed some of my concerns, but significant issues remain. I remain very concerned about the activation supposedly induced by changes in ionic strength. The authors now seem to be arguing that it is reduced divalent ion concentrations that activate the channel at normal ionic strength. So what is, reduced ionic strength, reduced divalents, a synergistic interaction between ionic strength and divalents or something else?

In our study, we were using an established protocol for channel activation to investigate the influence of the selected sybodies on VRAC function. The described protocol was derived from a classical paper that has demonstrated that the reduction of ionic strength is sufficient for the activation of VRAC channels¹. In our attempts to optimize this protocol, we found an effect of free Mg^{2+} in apparently inhibiting VRAC currents, which has also been described previously². This is consistent with our observation that the addition of Na^+ -ATP but not Mg^{2+} -ATP was required to increase current density. It thus appears that, besides exerting potential functional effects by either directly or indirectly interacting with the channel, unbound ATP might also buffer the free Mg^{2+} concentration. In this case, the low free Mg^{2+} concentration would act synergistically with the reduced ionic strength to activate VRAC channels. However, it should be pointed out that our work does neither aim to investigate the action of Mg^{2+} and ATP on VRAC activation nor do we want to engage in speculations on the physiological activation mechanism. Instead, our manuscript characterizes molecular interactions and their consequence on protein function where we use the applied protocol as a robust way to investigate the effect of sybody binding on channel activation.

We have previously recorded VRAC currents using a similar protocol in our article describing the LRRRC8A structure³. In our previous work, we showed activation of endogenous anion-selective currents in HEK293 cells by patch clamp electrophysiology in the whole-cell configuration in response to pipette solutions with reduced ionic strength that were osmotically balanced with the bath solution (see Extended Data Fig. 2a-c of the described article). This

response was absent in a cell-line where all five LRR subunits were knocked out (LRRC8^{-/-}, Extended Data Fig. 2d-e). Currents with similar phenotype as obtained for wildtype cells could be recovered after co-transfection of constructs coding for LRRC8A and C subunits (Fig. 1a, b, Extended Data Fig. 2f, g), whereas much smaller currents that required low salt concentrations for activation were observed for LRRC8A alone (Fig. 1a, c, Extended Data Fig. 2h, i). Since surface biotinylation showed targeting of the homomeric channel to the plasma membrane (Extended Data Fig. 2j) we concluded that the cause for the low current response is likely a consequence of a reduced single channel conductance and a low open probability, suggesting that the channel shows compromised activation properties. In the described study, we were also able to demonstrate the importance of an arginine (Arg 103) of LRRC8A that is located in the extracellular constriction of the pore for ion selectivity of heteromeric LRRC8A/C channels (Fig. 6e, f, Extended Data Fig. 9).

Whereas in our previous manuscript, most experiments were carried out at an intracellular salt concentration of 100 mM, we also observed a reduced, yet robust response already at 125 mM and thus decided to use this concentration for several experiments in our current work to explore the effect of sybody interaction on VRAC currents. In our study, we find a similar inhibitory effect of Sybody 1 (Sb1) on VRAC channels that are either activated by swelling (Supplementary Fig. 2) or by reduction of the ionic strength (Fig. 2a, c, Supplementary Fig. 3a, b). In contrast, no significant effect was observed upon expression of a control sybody that was generated to target a bacterial protein (Fig. 1c, Supplementary Fig. 2c, g). Strong inhibition of VRAC currents by Sb1 was also observed at lower ionic strength (Fig. 2c). The same protocol showed inhibition by Sb2 and Sb3 (Fig. 2c) and activation by Sb4 and Sb5, which becomes more pronounced at high intracellular ion concentration, a condition where endogenous channels are usually closed (Fig. 2e, f). Finally, we showed similar functional consequences of the respective sybodies upon interaction with homomeric LRRC8A channels expressed in LRRC8^{-/-} cells (Fig. 2g, h, Supplementary Fig. 3e, f). Together, these findings underline the similarity in the effect these sybodies exert on VRAC function irrespective of the activation mechanism and channel composition.

The suggestion that divalents play a regulatory role contradicts published studies. Multiple studies from several laboratories that are the leaders in the VRAC field have shown that

reducing ionic strength has two effects. First, as ionic strength is reduced, less swelling is required to activate the channel. Second, when ionic strength is reduced to grossly unphysiological levels, VRAC activates without swelling. All of this occurs in the presence of constant levels of divalent ions.

I do not see why our observation of a potential inhibitory role of Mg^{2+} would contradict published work since, except for the aforementioned study², I am not aware of any investigation that systematically addressed the role of Mg^{2+} on VRAC activation. Again, I want to emphasize that we do not want to engage in speculations on the physiological mechanism of VRAC activation as this is not the focus of the current study and we also do not claim that an effect of Mg^{2+} is of any physiological importance. However, it should be mentioned that the question of VRAC activation and the role of ionic strength appears to date to be still not fully clarified^{1,4-7}.

The presence of ATP in the cell is an absolute requirement for activation unless the rate of cell swelling is extremely high. The requirement for ATP occurs in the presence or absence of divalent ions and ATP can be replaced by nonhydrolyzable analogs. The authors present unpublished data in their rebuttal that contradict these peer-review published studies.

We do not question the role of ATP in activation by either direct or indirect interactions with the channel and want to emphasize that ATP was always present in our recording solutions. However, as mentioned above, we have found the effect of ATP to be strongly dependent on its counterion with Na^+ -ATP exerting a strong and Mg^{2+} -ATP a much smaller effect. This points towards a potential role for free ATP as chelator of Mg^{2+} , which in turn might increase the sensitivity of VRACs to mildly decreased ion concentrations.

To better illustrate the purpose of our electrophysiological recordings we have introduced the following changes to the manuscript.

Line 132-142:

‘HEK293 cells show a strong current response mediated by heteromeric channels of the LRRC8 family upon either cell swelling or the reduction of the intracellular ionic strength, although the relationship between both activation modes and the requirement of a certain degree of swelling

as a prerequisite for channel opening has remained controversial^{18,23,32,33}. We have previously used a protocol that relies on a reduced intracellular ionic strength in osmotically balanced conditions in combination with high ATP and low divalent ion concentrations, which synergistically lead to robust channel activation^{18,29} and employed this protocol in the present study (Fig. 2, Supplementary Figs. 2 and 3).’

Line 159-167:

In recordings measured under activating conditions, we did not observe any current response, irrespectively of whether activation proceeded by swelling using a previously described protocol⁸ or exposure of the cytoplasm to 125 mM salt (in conjunction with high ATP and low Ca²⁺ and Mg²⁺ concentrations, Fig. 2a, c, Supplementary Fig. 2a–c).

My criticism about drawing excessive conclusions from the LRRC8A structure remains. There can be no question that the LRRC8A homomer structure represents an important breakthrough in the field. However, we now know that LRRC8A homomers have grossly abnormal regulation and functional properties. Thus, drawing structural conclusions from LRRC8A homomers about how VRAC is regulated without molecular confirmation is unwarranted.

In our work, we try to refrain from drawing excessive conclusions from LRRC8A structures. Instead, we carefully describe the structural features of LRRC8A complexes in the ‘Results’ section and cautiously point towards potential functional implications in the ‘Discussion’. Although the fact that the sybodies exert similar functional consequences on homo- and heteromeric channels (Fig. 2, Supplementary Fig. 3) suggests that activation and inhibition likely follow similar general principles, there are certainly differences with respect to detailed mechanisms. As in our previous study³, we have described LRRC8A as channel with compromised activation properties, which is likely a consequence of the altered free energy difference between inactive and active conformations. This does not necessarily require grossly different structural features, which would be unlikely given the high sequence conservation between subunits. In fact, pronounced functional differences between homo- and heteromeric complexes have also been found for members of the pentameric ligand-gated ion channel family,

despite the general structural resemblance of the individual subunits and their assembly in the pentameric channel.

Although we expect distinct conformational properties in homo- and heteromeric channels that in a cellular context might also be stabilized by interactions with subunit-specific accessory proteins, we want to emphasize that we already find a large heterogeneity of conformations for LRRC8A. These heterogeneities are particularly noticeable on the level of the cytoplasmic domain, where even in the case of the LRRC8A or the LRRC8A-Sb1 complex, the described three-fold symmetric structure would only account for 1/4 to 1/3 of the classified particles, whereas other classes show a larger asymmetry of the domains (see Extended Data Figs. 3d, 4d from our previous study³ and Supplementary Fig. 4d in our current manuscript), a property that is even more pronounced in other complexes (Supplementary Figs. 5d, 6d, 7a, f and 8d). We have here focused on the description of the three-fold symmetric channels structure since it provides the most accurate picture of molecular interactions and since even structures displaying higher domain mobility show features of the tight interaction between LRR domain pairs described here. Although we do want to refrain from a definitive assignment of observed conformations to functional states, the results of our study hint towards domain rigidification being associated with low channel activity (as observed for the LRRC8A/Sb1 and Sb3 complexes, Supplementary Figs. 4 and 6), whereas structures in complex with activators display an increased domain mobility (Supplementary Figs. 7 and 8).

I don't believe the authors can state that the structural features of LRRC8A "are generally similar to VRAC heteromers" based on their work that has not yet been reproduced by other labs. The heteromeric structure generated by the Dutzler lab is low resolution, shows only a side view of the channel, and was reconstructed using the homomeric LRCC8A structure as a reference map and imposed C3 symmetry, both of which likely imposed a bias in the solved heteromeric structure. Furthermore, this low-resolution structure does not take into account the possibility that subunit stoichiometry could very well influence overall channel conformation.

Our statement on the general similarity between homo- and heteromeric channels refers to an 8 Å structure of a heteromeric channel composed of LRRC8A and C subunits described in our

previous manuscript³. The composition of the sample was investigated by mass spectrometry and revealed a comparable abundance of A and C subunits. This structure shows a channel with the same hexameric assembly, similar overall dimensions and an equivalent arrangement of membrane-spanning helices as found for LRRC8A (Fig. 1d, e, Extended Data Fig. 5). Although the conformational flexibility of the cytosolic domains was increased compared to the LRRC8A homomer, a population of the channel showed a comparable domain arrangement as seen in the C3-symmetric structure of LRRC8A. Due to their strong conservation, we expect the A and C subunits in the hexameric protein to be averaged. At this stage, our comparison is limited to these low-resolution features, and we thus refer to a general similarity of the overall structures. We do not want to imply that there were no differences between homo- and heteromeric channels at higher resolution, including specific conformational preferences, which might underly the distinct activation properties of either channel.

The fact that comparable structures of heteromeric LRRC8 channels have not yet been reported by other labs might be related to the considerable experimental efforts associated with the expression and purification of such heteromeric channels for which we usually pool cells from 10-15 l of suspension culture. It should also be emphasized that the described structure only provides a first and preliminary view of a heteromeric channel at low resolution, which will have to be improved in future investigations. Since we are interested in the architecture of heteromeric LRRC8A channels, we have repeated similar experiments several times with equivalent results. The described map has been deposited in the EMD and can be downloaded for inspection. The structure determination process is described in detail in Extended Data Fig. 5 of our previous study³ and illustrated in Fig. 1 below to rebut claims that the observed structure would be a model artefact.

The danger of model bias in this structure is unsubstantiated since the LRRC8A map used as initial reference for the assignment of Euler angles was low-pass filtered at 40 Å and thus merely resembles a blob of approximate shape of the protein (Response Fig. 1a). All molecular features described in the model emerged during refinement in RELION, whose maximum likelihood approach further minimizes bias (Response Fig. 1c-f). In fact, the classification and 3D refinement were carried out both in the absence or presence of symmetry with similar general results (Extended Data Fig. 5f of our previous study³ and Response Fig. 1e, f).

To better emphasize our cautious interpretation of structural data, we have introduced the following changes to our manuscript:

Abstract. Line 17-21:

‘We used these binders to investigate their interaction with homomeric LRRC8A channels by cryo-electron microscopy and the consequent effect on channel activation by electrophysiology. The five identified sybodies either inhibit or enhance activity by binding to distinct epitopes of the LRR domain, thereby altering channel conformations.’

Introduction, line 47-57:

‘With respect to their structure, homomeric LRRC8A channels appear to exhibit general features that are also observed in heteromeric proteins¹⁸. This assumption is based on a low-resolution structure obtained from a preparation of LRRC8A oligomers containing A and C subunits and refers to the hexameric organization of channels and their general structural features whereas differences in the molecular details and distinct conformational properties are expected to persist between homo- and heteromers. LRRC8 channels share a modular organization consisting of a membrane-inserted pore domain and cytoplasmic leucine-rich repeat (LRR) domains¹⁸.’

Introduction, line 68-70:

‘The functional relationship between the C3-symmetric channel structure and conformations with asymmetric LRR domain arrangement is still unknown.’

Results, line 209-214:

‘A large population of the particles (i.e. 26% of the particles used for 3D classification) shows a similar C3-symmetric structural arrangement as previously observed for the apo-protein (Fig. 4a–c, Supplementary Fig. 4d, e). Other classes (in total encompassing 74% of the classified particles) show a well-defined pore domain but different degree of mobility of the cytoplasmic LRR domains). In the C3-symmetric structure, the densities of sybodies define the interaction of the binder with the channel at the lower part of the cytoplasmic domain towards the intracellular side (Fig. 3a, Supplementary Fig. 10a, b).’

Discussion, line 371-381:

‘The respective complex structures thus likely display general properties of sybody interactions that might also extend towards heteromeric channels. Since the local concentration of the targeted A-subunit is increased compared to their heteromeric equivalents, we assume observed

structural features to be even enhanced in homomeric channels. However, due to the unknown disposition of subunits in LRRC8 heteromers, we also expect unique properties of subunit interactions in endogenous heteromeric channels, which will have to be explored in future studies.'

Discussion, line 389-407:

'The large number of ionizable residues suggest a plausible dependence of interactions between LRR domains on the ionic strength that could be weakened by the shielding of charges at higher salt concentrations and a hypothetical interaction with divalent cations, although the role of such interactions will have to be clarified in future studies (Supplementary Fig. 11c-e).'

Response Fig. 1 Structural features of a heteromeric LRRC8 channel composed of A and C subunits. **a**, 40 Å low-pass filtered map of LRRC8A used as initial template for the assignment of Euler angles of LRRC8C/D particles. **b-d**, C3-symmetrized maps of homomeric LRRC8A channels at low contour (4σ) low-pass filtered at 8 Å and the LRRC8A/C channel at the same resolution. Views are from **b**, within the membrane, **c** the extracellular side and **d**, the cytoplasm. Panels show, left, LRRC8A (grey), center, LRRC8A/C (yellow) and right, a superposition of both maps. **e, f** Comparison of C3-symmetrized and unsymmetrized (C1) maps of LRRC8A/C to the C3-symmetrized map of LRRC8A at 8 Å contoured at 10.5σ . **e**, Channel viewed from within the membrane. **f**, Slice of the membrane-inserted part of the pore viewed from the extracellular side. **e, f**, The C3-symmetrized map of LRRC8A is shown in grey, the C3 symmetrized map of LRRC8A/C in yellow and the non-symmetrized (C1) map in cyan. ‘superimposed’ refers to a superposition of respective maps indicated by their coloring.

References

- 1 Voets, T., Droogmans, G., Raskin, G., Eggermont, J. & Nilius, B. Reduced intracellular ionic strength as the initial trigger for activation of endothelial volume-regulated anion channels. *Proc Natl Acad Sci U S A* **96**, 5298-5303, doi:10.1073/pnas.96.9.5298 (1999).
- 2 Lazarenko, P. M., Pohoriela, N. & Shuba Ia, M. [Adenosine triphosphate-dependence of volume sensitive chloride current in LNCaP cell line of human prostate cancer]. *Fiziol Zh* **51**, 51-61 (2005).
- 3 Deneka, D., Sawicka, M., Lam, A. K. M., Paulino, C. & Dutzler, R. Structure of a volume-regulated anion channel of the LRRC8 family. *Nature* **558**, 254-259, doi:10.1038/s41586-018-0134-y (2018).
- 4 Cannon, C. L., Basavappa, S. & Strange, K. Intracellular ionic strength regulates the volume sensitivity of a swelling-activated anion channel. *Am J Physiol* **275**, C416-422, doi:10.1152/ajpcell.1998.275.2.C416 (1998).
- 5 Nilius, B., Prenen, J., Voets, T., Eggermont, J. & Droogmans, G. Activation of volume-regulated chloride currents by reduction of intracellular ionic strength in bovine endothelial cells. *J Physiol* **506 (Pt 2)**, 353-361, doi:10.1111/j.1469-7793.1998.353bw.x (1998).
- 6 Syeda, R. *et al.* LRRC8 Proteins Form Volume-Regulated Anion Channels that Sense Ionic Strength. *Cell* **164**, 499-511, doi:10.1016/j.cell.2015.12.031 (2016).
- 7 Pedersen, S. F., Okada, Y. & Nilius, B. Biophysics and Physiology of the Volume-Regulated Anion Channel (VRAC)/Volume-Sensitive Outwardly Rectifying Anion Channel (VSOR). *Pflugers Arch* **468**, 371-383, doi:10.1007/s00424-015-1781-6 (2016).